# KEEPLORA: CONTINUAL LEARNING WITH RESIDUAL GRADIENT ADAPTATION

**Mao-Lin Luo**[1,2], **Zi-Hao Zhou**[1,2], **Yi-Lin Zhang**[1,2], **Yuanyu Wan**[3], **Min-Ling Zhang**[1,2],
**Tong Wei**[1,2†]

[1]School of Computer Science and Engineering, Southeast University, Nanjing 210096, China
[2]Key Laboratory of Computer Network and Information Integration (Southeast University),
 Ministry of Education, China
[3]School of Software Technology, Zhejiang University, Ningbo, China

## ABSTRACT

Continual learning for pre-trained vision-language models requires balancing three competing objectives: retaining pre-trained knowledge, preserving knowledge from a sequence of learned tasks, and maintaining the plasticity to acquire new knowledge. This paper presents a simple but effective approach called *KeepLoRA* to effectively balance these objectives. We first analyze the knowledge retention mechanism within the model parameter space and find that general knowledge is mainly encoded in the *principal* subspace, while task-specific knowledge is encoded in the *residual* subspace. Motivated by this finding, KeepLoRA learns new tasks by restricting LoRA parameter updates in the residual subspace to prevent interfering with previously learned capabilities. Specifically, we infuse knowledge for a new task by projecting its gradient onto a subspace orthogonal to both the principal subspace of pre-trained model and the dominant directions of previous task features. Our theoretical and empirical analyses confirm that KeepLoRA balances these three objectives and achieves state-of-the-art performance. The implementation code is available at https://github.com/MaolinLuo/KeepLoRA.

## 1 INTRODUCTION

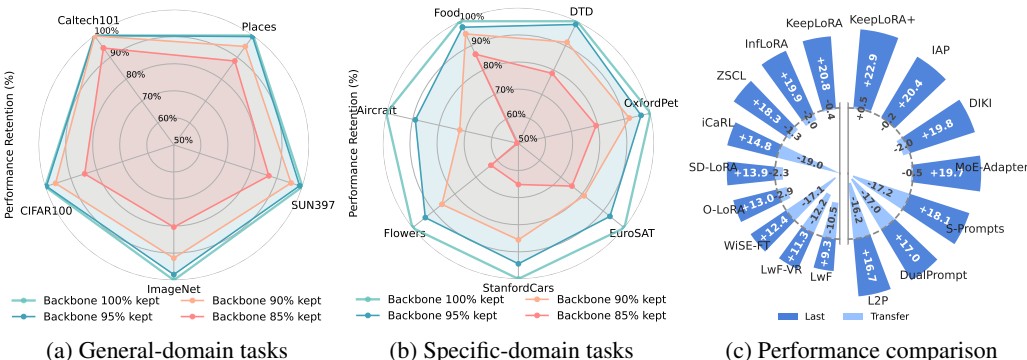

(a) General-domain tasks   (b) Specific-domain tasks   (c) Performance comparison

Figure 1: Analysis of model parameter subspaces and overall CL performance. In Fig. 1a and 1b, we measure zero-shot performance after reconstructing attention weights using only the top principal singular components. While performance on general-domain datasets remains highly robust, performance on most specific-domain datasets degrades sharply as more low-energy components are removed. In Fig. 1c, the *Last* metric measures the accuracy gain on the final learned task relative to a zero-shot baseline, while *Transfer* measures the accuracy degradation on unseen tasks.

---

[†]Corresponding author.

Table 1: Comparison of LoRA-based approaches to continual learning. When learning the $t$-th task, O-LoRA regularizes the down-projection matrix to be orthogonal to that of previous tasks to improve stability; InfLoRA constrains the optimization of task features $H_t$ to be orthogonal to previous dominant directions $M_{t-1}$ to jointly improve plasticity and stability; SD-LoRA optimizes a decoupled LoRA to improve stability and re-scales the magnitudes $\{\alpha_i\}_{i=1}^{t-1}$ of parameters from previous tasks to improve plasticity. By contrast, our method constrains optimization to a subspace orthogonal to both the principal weight subspace $W_p$ and previous task directions $M_{t-1}$ to preserve stability, and initializes the update in an optimal gradient space derived from $G_t$ to boost plasticity.

| Method | Initialization | | Training Objective |
|---|---|---|---|
| | LoRA $A$ | LoRA $B$ | |
| O-LoRA (Wang et al., 2023a) | $A \leftarrow \mathcal{N}(0, \sigma^2)$ | $B \leftarrow 0$ | $\mathcal{L}_{\text{cls}}(A_t, B_t) + \sum_{i=0}^{t-1} \|A_t^\top A_i\|_F^2$ |
| InfLoRA (Liang & Li, 2024) | $USV^\top = \text{SVD}\left(H_t - M_{t-1}M_{t-1}^\top H_t\right)$ $A \leftarrow U_r$ | $B \leftarrow 0$ | $\mathcal{L}_{\text{cls}}(B_t)$ |
| SD-LoRA (Wu et al., 2025b) | $A \leftarrow \mathcal{N}(0, \sigma^2)$ | $B \leftarrow 0$ | $\mathcal{L}_{\text{cls}}(\{\alpha_i\}_{i=1}^{t-1}, \alpha_t \overline{A_t B_t})$ |
| KeepLoRA (This paper) | $\text{SVD}(G_t - W_p W_p^\top G_t - M_{t-1}M_{t-1}^\top G_t)$ $A \leftarrow U_r$ | $B \leftarrow S_r V_r^\top$ | $\mathcal{L}_{\text{cls}}(B_t)$ |

Vision-language models (VLMs) have demonstrated remarkable zero-shot transfer capabilities, making them cornerstones of many downstream applications (Comanici et al., 2025; Achiam et al., 2023; Radford et al., 2021). Despite this success, their performance on certain datasets can be insufficient, motivating the need for continual learning (CL). An effective CL method requires the balance of three competing objectives: maintaining the ability to learn new knowledge (*plasticity*), preventing the forgetting of previously learned tasks (*backward stability*), and crucially, preserving the general pre-trained knowledge that guarantees general transferability (*forward stability*) (Mukhoti et al., 2024; Zheng et al., 2023). Degradation of this pre-trained knowledge is particularly detrimental, as it erodes the core value of VLMs. Therefore, the central challenge is how to effectively learn new knowledge without undermining these critical stability constraints.

A straightforward solution, replaying pre-training data, is rarely viable due to prohibitive computational costs and the frequent unavailability of proprietary training corpora (Wang et al., 2024; Zhou et al., 2024; Rolnick et al., 2019). The current alternatives largely follow two paths. The first is (i) *reference-data regularization*, which uses reference data to anchor the model parameters and retain stability (Wu et al., 2025a; Zheng et al., 2023). However, the success of these approaches is highly sensitive to the choice of reference data with additional training costs (Luo et al., 2025; Zheng et al., 2023). The second path involves (ii) *architecture extension*, such as prompt-pool (Fu et al., 2025; Wang et al., 2022b) or MoE-adapters (Yu et al., 2024; Dou et al., 2024) while freezing the model backbone. Although effective in preventing forgetting, these modules increase inference costs (Nayak et al., 2025) and complicate the deployment (Cai et al., 2025; Zadouri et al., 2023). Since trained weights are compact representations of data (Deletang et al., 2024; Franceschelli et al., 2024), an ideal CL method should infuse new knowledge directly into the existing parameter space, leveraging its inherent redundancy rather than accumulating external modules (Sharma et al., 2024).

To identify where new knowledge can be infused without disrupting existing abilities, we analyze the parameter space of the backbone attention weights via singular value decomposition (SVD). As shown in Fig. 1a and 1b, our analysis reveals that the principal subspace, spanned by components with large singular values, predominantly encodes general knowledge, while the residual subspace, associated with small singular values, encodes domain-specific knowledge. This observation indicates that performance on specialized datasets is highly sensitive to alterations in the residual subspace, whereas general datasets remain robust to such changes. This insight forms the basis of our approach: to coherently achieve *forward stability*, *backward stability*, and *plasticity*, CL updates should be constrained to the residual subspace, enabling the acquisition of new domain-specific knowledge without affecting the principal subspace that encodes general knowledge.

We implement our subspace-constrained updates using low-rank adaptation, a parameter-efficient method whose updates can be merged into the original weights post-training, thus incurring no inference overhead. Existing low-rank CL methods, such as O-LoRA (Wang et al., 2023a), InfLoRA (Liang & Li, 2024), and SD-LoRA (Wu et al., 2025b), lack consideration of the transfer capacity of

the pre-trained model and update directions within suboptimal subspaces, which limit their plasticity and stability as shown in Tab. 1. To overcome this, we initialize the low-rank update using the gradient from the first training step. This ensures that the update direction closely approximates the full-parameter tuning gradient to boost plasticity. To keep stability, we explicitly project this update into the residual subspace of the pre-trained weights, which constrains learning on directions that do not interfere with the model's core transferable knowledge. Building on this, we construct a unified subspace that stores both the principal subspace of the model parameters and the dominant feature directions of each learned task. The unified principal subspace, with a size not exceeding the square of feature dimension, preserves both pre-trained and newly acquired knowledge.

Our contributions are summarized as follows:

- We empirically analyze the parameter space of pre-trained models and find that general knowledge is primarily encoded in the parameter principal subspace, while domain-specific adaptations are better captured by the residual subspace.

- We propose *KeepLoRA*, a novel method that leverages residual subspace constraints for parameter updates, validated by theoretical analysis showing how it optimally balances plasticity and stability through orthogonal projections.

- Experiments on dual-encoder (CLIP) and encoder-decoder (LLaVA) models validate that KeepLoRA effectively balances the three core challenges of plasticity, backward stability, and forward stability, establishing new state-of-the-art results on benchmark datasets.

## 2 RELATED WORKS

In continual learning, forward stability is typically preserved by using reference-data regularization and architecture extension techniques. In addition, gradient projection methods are commonly employed to address backward stability and plasticity. In this section, we review these lines of work.

**Reference-Data Regularization.** Continual learning on narrow task distributions can cause the model feature space to collapse, degrading its pre-trained zero-shot transfer capabilities (Zheng et al., 2023). Reference-data methods aim to counteract this by anchoring the model representations. ZSCL (Zheng et al., 2023) uses the ImageNet (Deng et al., 2009) and Conceptual Captions (Sharma et al., 2018) datasets as reference data, employing distillation to preserve the structure of the feature space. However, the effectiveness of this approach is sensitive to the choice of reference data and the teacher model, with performance degrading when fewer images or classes are used (Zheng et al., 2023). Yu et al. (2024) propose MoE-Adapters by training a selector on the TinyImageNet (Deng et al., 2009) dataset to identify out-of-distribution data, which is then processed by the original frozen model. Wu et al. (2025a) leverage the generative model Stable Diffusion (Rombach et al., 2022) to create synthetic reference data for distillation. These methods inherently increase computational overhead and depend on external reference data, limiting their practical feasibility.

**Architecture Extension.** Architecture extension methods freeze the pre-trained model and extend it with new parameters for each task. L2P (Wang et al., 2022c) selects the most relevant prompts from a prompt pool, while DualPrompt (Wang et al., 2022b) uses task-sharing and task-specific prompts. CODA-Prompt (Smith et al., 2023) proposes end-to-end prompt selection to increase plasticity. MoE-Adapters (Yu et al., 2024) inserts a mixture of adapters into the image encoder, activating a subset for each task. DIKI (Tang et al., 2024) calibrates knowledge integration by determining the likelihood that a test sample belongs to a task. IAP (Fu et al., 2025) introduces Instance-Aware Gated Prompting to improve the effectiveness of prompt selection. However, these methods often suffer from parameter selection errors or suboptimal activation coefficients. Moreover, adding external parameters does not truly infuse new knowledge into the base model.

**Gradient Projection.** Gradient projection methods mitigate catastrophic forgetting by constraining parameter updates into specific subspaces, thereby preventing interference with previously acquired knowledge (Qiao et al., 2024). In the context of full fine-tuning, methods such as Gradient Projection Memory (GPM) (Saha et al., 2021) enforce orthogonality between the gradients of a new task and a stored basis of principal gradient directions from previous tasks. To improve the efficiency of full fine-tuning, CoSo (Cheng et al., 2025) utilizes Task-Specific Subspace Estimation and updates an orthogonal basis matrix. This thought has also been adapted to parameter-efficient techniques. For

example, O-LoRA (Wang et al., 2023a) constrains the LoRA subspaces of new tasks to be orthogonal to those of previous tasks, ensuring that learning occurs in novel directions. InfLoRA (Liang & Li, 2024) applies a constraint where the LoRA down-projection matrix $\boldsymbol{A}$ is orthogonal to GPM (Saha et al., 2021) or DualGPM (Liang & Li, 2023) to prevent interference. However, these existing methods primarily focus on mitigating backward forgetting, the loss of knowledge from previously learned sequential tasks. They do not explicitly address the preservation of pre-trained knowledge, which is crucial for maintaining the model's transferability and preventing forward forgetting.

## 3 METHOD

### 3.1 PRELIMINARY

**Problem Formulation.** We adopt the multi-domain task incremental learning (MTIL) setting (Zheng et al., 2023), where the model encounters a sequence of $n$ tasks $\{\mathcal{T}^1, \mathcal{T}^2, \ldots, \mathcal{T}^n\}$. Each task $\mathcal{T}^i = (\mathcal{D}^i, \mathcal{C}^i)$ for $i \in \{1, 2, \ldots, n\}$ comprises a dataset $\mathcal{D}^i$ and corresponding class vocabulary $\mathcal{C}^i$. The dataset $\mathcal{D}^i = \{(\boldsymbol{x}_j^i, y_j^i)\}_{j=1}^{N_i}$ contains $N_i$ training examples, where each $\boldsymbol{x}_j^i$ denotes an input image and $y_j^i$ represents the corresponding one-hot encoded ground truth label. The class vocabulary $\mathcal{C}^i = \{c_j^i\}_{j=1}^{m_i}$ establishes the mapping between categorical labels and semantic class names, with $m_i$ denoting the total number of distinct classes for task $\mathcal{T}^i$. During inference, the model classifies an input image $\boldsymbol{x}$ within $\mathcal{C}^i$. The goal of continual learning is to maintain performance on pre-trained knowledge and all previously encountered tasks while adapting to new ones.

**Vanilla LoRA.** Low-rank adaptation (LoRA) (Hu et al., 2022) decomposes weight updates into two low-rank matrices $\boldsymbol{A} \in \mathbb{R}^{d_{\text{in}} \times r}$ and $\boldsymbol{B} \in \mathbb{R}^{r \times d_{\text{out}}}$, where $r \ll \min(d_{\text{in}}, d_{\text{out}})$. During training, $\boldsymbol{W}$ remains frozen while only $\boldsymbol{A}$ and $\boldsymbol{B}$ are fine-tuned. The matrices are initialized with $\boldsymbol{A} \sim \mathcal{N}(0, \sigma^2)$ and $\boldsymbol{B} = \boldsymbol{0}$. For input $\boldsymbol{x} \in \mathbb{R}^{d_{\text{in}}}$, the forward pass becomes:

$$y = x \left( W + \frac{\alpha}{r} AB \right) \tag{1}$$

where $\alpha$ is a scaling factor.

### 3.2 KEEPLORA: GRADIENT PROJECTION ADAPTATION

Continual learning for pre-trained vision-language models demands a balance between *plasticity*, the ability to acquire new knowledge, and *learning stability*, which comprises both *forward stability* to preserve general pre-trained knowledge and *backward stability* to retain knowledge from previously learned tasks. To address this problem, we propose *KeepLoRA*, a method built upon LoRA that employs residual subspace constraints to unify stability preservation and new knowledge infusion.

**Stability: Preserving Pre-trained and Previous Task Knowledge.** KeepLoRA retains stability by projecting the subspaces of pre-trained knowledge and previous task knowledge onto a unified principal subspace. Subsequent adaptations for new tasks are then confined to the residual subspace orthogonal to this principal subspace, thereby minimizing interference with the learned knowledge.

*Pre-trained Knowledge Subspace:* We analyze the parameters of the pre-trained model to understand how the model stores general knowledge. Specifically, we decompose each weight matrix $\boldsymbol{W} \in \mathbb{R}^{d_{in} \times d_{out}}$ as $\boldsymbol{W} = \boldsymbol{U}\boldsymbol{S}\boldsymbol{V}^\top$. The decomposition produces a subspace $\boldsymbol{W}_p = \boldsymbol{U}_{:,1:p}$, and the subspace is constrained such that:

$$||\boldsymbol{W}_p||_F^2 \geq \epsilon_w ||\boldsymbol{W}||_F^2 \tag{2}$$

where $\epsilon_w \in (0, 1)$ controls the energy ratio retained in $\boldsymbol{W}_p$.

*Previous Task Knowledge Subspace:* To mitigate forgetting of learned tasks, the LoRA module updating matrix $\boldsymbol{W}$ for new tasks should minimize interference with features from previous tasks. Specifically, our goal is to make $\boldsymbol{Y} = \text{LoRA}_t(\boldsymbol{X})$ as close to $\boldsymbol{0}$ as possible for any input $X$ from previous tasks $\{\mathcal{T}_i\}_{i=1}^{t-1}$. Since no real or synthetic samples from previous tasks are available for replay, we propose to extract the dominant singular vectors of previous tasks as the dominant feature directions. This approach enables us to continuously compress task-specific information and enforce matrix $\boldsymbol{A}$ to be orthogonal to the dominant singular vectors on LoRA initialization. After training for task $t$, we extract and store the dominant feature directions for this task. These directions are

chosen to be orthogonal to the subspace jointly defined by the principal weights and the dominant feature directions of all $t-1$ tasks. We then define the feature space for the $t$-th task as:

$$\hat{\boldsymbol{X}}_t = \boldsymbol{X}_t - \boldsymbol{W}_p\boldsymbol{W}_p^\top\boldsymbol{X}_t - \boldsymbol{M}_{t-1}\boldsymbol{M}_{t-1}^\top\boldsymbol{X}_t \tag{3}$$

where $\boldsymbol{M}_{t-1} \in \mathbb{R}^{d_{in}\times k}$ represents the accumulated direction matrix containing the dominant singular vectors from tasks $\{1, 2, \ldots, t-1\}$, and $k$ denotes the total number of stored singular vectors. We initialize $\boldsymbol{M}_0 = \emptyset$ as an empty matrix. The number of stored vectors $k$ is dynamically determined by an energy threshold $\epsilon_f \in (0, 1)$. Specifically, we retain the minimum number $k$ of dominant directions required to satisfy:

$$||\hat{\boldsymbol{X}}_t||_F^2 + ||\boldsymbol{W}_p\boldsymbol{W}_p^\top\boldsymbol{X}_t||_F^2 + ||\boldsymbol{M}_{t-1}\boldsymbol{M}_{t-1}^\top\boldsymbol{X}_t||_F^2 \geq \epsilon_f||\boldsymbol{X}_t||_F^2 \tag{4}$$

We perform SVD on the features $\hat{\boldsymbol{X}}_t = \boldsymbol{U}_t\boldsymbol{S}_t\boldsymbol{V}_t^\top$ and extract the top-$m$ dominant singular vectors to update our subspace matrix: $\boldsymbol{M}_t = [\boldsymbol{M}_{t-1}, \boldsymbol{V}_{t(:,1:m)}]$, where $m$ is determined by a threshold $\epsilon_f$.

*Unified Principal Subspace.* Since both $\boldsymbol{W}_p$ and $\boldsymbol{M}_t$ consist of orthogonal direction vectors operating within the same $d_{in}$-dimensional feature space, and the total number of orthogonal vectors is upper-bounded by $d_{in}$, we can mathematically unify them into a single projection subspace: $\boldsymbol{M}_t' = [\boldsymbol{W}_p, \boldsymbol{M}_t]$. The unified subspace leverages the theoretical foundation that predictive models can be transformed into lossless compressors (Deletang et al., 2024) and model weights embody a compressed representation of the training data (Franceschelli et al., 2024). Under this perspective, $\boldsymbol{W}_p$ captures the essential feature representation space of the pre-training data, while $\boldsymbol{M}_t$ preserves the dominant feature directions during continual learning. Both components represent compressed knowledge from their respective data distributions.

To ensure the new $t$-th task updates never interfere with $\boldsymbol{M}_{t-1}'$, KeepLoRA achieves this through a modified LoRA approach, where matrix $\boldsymbol{A}$ is initialized within $\{\boldsymbol{M}_{t-1}'\}^\perp$ and frozen throughout training, while only $\boldsymbol{B}$ is optimized.

**Plasticity: Gradient-Informed LoRA Initialization in Residual Subspace.** While the unified principal subspace ensures learning stability, KeepLoRA also requires maintaining plasticity to adapt to new tasks. We achieve it by initializing the LoRA module using task-specific gradient information, aligning adaptation directions with full fine-tuning while confining updates to $\{\boldsymbol{M}_{t-1}'\}^\perp$. Specially, we utilize gradient information to guide the initialization within the constrained residual space. Let $\boldsymbol{G}_t = \nabla_{\boldsymbol{W}}\mathcal{L}(\boldsymbol{W}; \mathcal{D}^t)$ denotes the gradient of the weight matrix $\boldsymbol{W}$ of the $t$-th task at the first training step. We project this gradient onto the residual subspace:

$$\hat{\boldsymbol{G}}_t = \underbrace{\boldsymbol{G}_t}_{\text{plasticity}} - \underbrace{\boldsymbol{W}_p\boldsymbol{W}_p^\top\boldsymbol{G}_t - \boldsymbol{M}_{t-1}\boldsymbol{M}_{t-1}^\top\boldsymbol{G}_t}_{\text{forward and backward stability}} \tag{5}$$

We perform SVD on the projected gradient $\hat{\boldsymbol{G}} = \boldsymbol{U}\boldsymbol{S}\boldsymbol{V}^\top$ and initialize the LoRA matrices with top-$r$ singular vectors as:

$$\boldsymbol{A} = \boldsymbol{U}_{:,1:r}, \quad \boldsymbol{B} = \boldsymbol{S}_{1:r}\boldsymbol{V}_{:,1:r}^\top \tag{6}$$

where $\boldsymbol{U}_{:,1:r}$ denotes the first $r$ columns of $\boldsymbol{U}$, and $r$ is the rank parameter. This gradient-informed initialization directly simulates the update direction of full fine-tuning while operating within the residual subspace, enabling effective adaptation without undermining these critical stability constraints. Since the initial product $\frac{\alpha}{r}\boldsymbol{A}\boldsymbol{B}$ is non-zero, the frozen parameter $\boldsymbol{W}$ can be adjusted to maintain the initial parameter values unchanged. Specifically, we replace the original parameter $\boldsymbol{W}$ with $\boldsymbol{W}' = \boldsymbol{W} - \frac{\alpha}{r}\boldsymbol{A}\boldsymbol{B}$ to ensure that the initial forward pass behavior remains identical with the initial model. Algorithm 1 summarizes the proposed KeepLoRA method.

## 3.3 DISCUSSION OF KEEPLORA

Eq. 5 and Eq. 6 serve as the core formulas of KeepLoRA, enabling its balance of plasticity and stability: $\boldsymbol{G}_t$ enhances plasticity by identifying new task adaptation directions, while the subtracted terms remove gradients that interfere with pre-trained and previous task knowledge, ensuring stability. To verify these core designs, we first establish the equivalence between KeepLoRA parameter update rule and gradient projection learning, defining the necessary properties of the subspace spanned by $\boldsymbol{A}_t$. We then demonstrate that the initialization of $\boldsymbol{A}_t$ meets these properties.

---

**Algorithm 1** KeepLoRA for Continual Learning

---

1: **Input:** Pre-trained model $f_\theta$ with updatable parameters $\{\boldsymbol{B}_i\}$, task sequence $\{\mathcal{T}^t\}_{t=1}^n$, hyper-parameters $\epsilon_w, \epsilon_f, r, \alpha$
2: **Output:** Updated model $f_{\theta'}$ with merged LoRA adapters
3: **for** task $t = 1$ to $n$ **do**
4:     Initialize KeepLoRA through Eq. 5 and Eq. 6
5:     Replace the parameter $\boldsymbol{W}$ with the modified frozen parameter $\boldsymbol{W}' = \boldsymbol{W} - \frac{\alpha}{r}\boldsymbol{A}_t\boldsymbol{B}_t$
6:     Compute the loss and optimize the KeepLoRA parameters $\boldsymbol{B}_t$
7:     Merge KeepLoRA and current model by $\boldsymbol{W} = \boldsymbol{W}' + \frac{\alpha}{r}\boldsymbol{A}_t\boldsymbol{B}_t$
8:     Extract dominant feature directions through Eq. 3 and Eq. 4
9: **end for**

---

**Analyzes of Frozen $\boldsymbol{A}_t$ LoRA Updates.** The KeepLoRA parameter update method involves freezing $\boldsymbol{A}_t$ and optimizing only $\boldsymbol{B}_t$. The following proposition demonstrates that this update rule is equivalent to gradient descent constrained within the subspace span($\boldsymbol{A}_t$).

**Proposition 3.1.** *(LoRA with frozen down-projection $\boldsymbol{A}_t$ is equivalent to gradient projection update.) Let $\mathcal{L}(\boldsymbol{W}; \mathcal{D}^t)$ denote the loss function for the t-th task $\mathcal{T}^t$, where: $\boldsymbol{W} = \boldsymbol{W}' + \frac{\alpha}{r}\boldsymbol{A}_t\boldsymbol{B}_t$, $\boldsymbol{G}_t = \nabla_{\boldsymbol{W}}\mathcal{L}(\boldsymbol{W}; \mathcal{D}^t)$. Optimizing only $\boldsymbol{B}_t$ through gradient descent with learning rate $\eta$ is equivalent to performing gradient descent on the orthogonal projection of $\boldsymbol{G}_t$ onto span($\boldsymbol{A}_t$). The weight update of $\boldsymbol{W}$ satisfies:*

$$\Delta\boldsymbol{W} = \frac{\alpha}{r}\boldsymbol{A}_t\Delta\boldsymbol{B}_t = -c\boldsymbol{A}_t\boldsymbol{A}_t^\top\boldsymbol{G}_t, \tag{7}$$

*where $c = \frac{\eta\alpha^2}{r^2}$ is a positive constant integrating the learning rate and LoRA scaling effects.*

**Remark.** Proposition 3.1 reveals that frozen $\boldsymbol{A}_t$ updates are inherently subspace constrained: all changes to $\boldsymbol{W}$ are confined to span($\boldsymbol{A}_t$), as $\boldsymbol{A}_t\boldsymbol{A}_t^\top$ acts as an orthogonal projection operator on this subspace. Furthermore, span($\boldsymbol{A}_t$) is required to satisfy the following two properties in continual learning: (i) Orthogonal to knowledge subspaces: span($\boldsymbol{A}_t$) needs to be orthogonal to subspaces encoding pre-trained knowledge and previously learned tasks, ensuring updates to $\boldsymbol{W}$ do not interfere with existing knowledge, preventing both forward and backward forgetting. (ii) Adaptation to the current task: span($\boldsymbol{A}_t$) needs to capture the dominant directions of $\boldsymbol{G}_t$, approximating the gradient of full-parameter fine-tuning to maintain plasticity.

**Validation of KeepLoRA $\boldsymbol{A}_t$ Initialization.** The preceding proposition outlines the required properties of span($\boldsymbol{A}_t$). The key question is whether the KeepLoRA initialization of $\boldsymbol{A}_t$ meets the two properties. We validate it by connecting the initialization to a constrained optimization problem.

**Proposition 3.2.** *KeepLoRA initialization of $\boldsymbol{A}_t$ through Eq. 5 and Eq. 6 is the solution to the following constrained optimization problem:*

$$\min_{\boldsymbol{A}_t^\top\boldsymbol{A}_t=\boldsymbol{I}} \|\boldsymbol{G}_t - \boldsymbol{A}_t\boldsymbol{A}_t^\top\boldsymbol{G}_t\|_F^2,$$
$$s.t \quad \boldsymbol{W}_p^\top\boldsymbol{A}_t = \boldsymbol{M}_{t-1}^\top\boldsymbol{A}_t = \boldsymbol{0}, \tag{8}$$

*where $\boldsymbol{G}_t$ is the current task gradient w.r.t. the base model $\boldsymbol{W}$, $\boldsymbol{W}_p$ is the principal subspace of pre-trained parameters, and $\boldsymbol{M}_{t-1}$ is the dominant feature directions from previous tasks.*

**Remark.** Proposition 3.2 directly connects KeepLoRA's initialization technique to the two properties of Proposition 3.1, verifying its optimality: (i) Satisfying orthogonality (via constraints): The equality constraints $\boldsymbol{W}_p^\top\boldsymbol{A}_t = \boldsymbol{0}$ and $\boldsymbol{M}_{t-1}^\top\boldsymbol{A}_t = \boldsymbol{0}$ explicitly enforce span($\boldsymbol{A}_t$) $\perp$ span($\boldsymbol{W}_p$) and span($\boldsymbol{A}_t$) $\perp$ span($\boldsymbol{M}_{t-1}$). It guarantees that span($\boldsymbol{A}_t$) is orthogonal to both the principal subspace of the model parameters and the dominant feature directions to preserve stability. (ii) Optimal adaptation (via objective): The objective function minimizes the Frobenius norm of $\boldsymbol{G}_t - \boldsymbol{A}_t\boldsymbol{A}_t^\top\boldsymbol{G}_t$, the residual component of $\boldsymbol{G}_t$ that lies outside span($\boldsymbol{A}_t$). By the Pythagorean theorem for the Frobenius norms ($\|\boldsymbol{G}_t\|_F^2 = \|\boldsymbol{A}_t\boldsymbol{A}_t^\top\boldsymbol{G}_t\|_F^2 + \|\boldsymbol{G}_t - \boldsymbol{A}_t\boldsymbol{A}_t^\top\boldsymbol{G}_t\|_F^2$), minimizing this residual is equivalent to maximizing the norm of the projected gradient $\boldsymbol{A}_t\boldsymbol{A}_t^\top\boldsymbol{G}_t$. It ensures span($\boldsymbol{A}_t$) captures the dominant gradient directions for the current task, preserving plasticity.

In summary, Propositions 3.1 and 3.2 form a complete theoretical loop: Proposition 3.1 defines the necessary properties of span($\boldsymbol{A}_t$) for stable-plastic continual learning. Proposition 3.2 proves

Table 2: Comparison of different methods on MTIL for each **classification** task in terms of *Transfer*, *Average*, and *Last* scores (%). The best results are in **bold**.

| Method | Arch. Kept | w/o Extra Data | Aircraft | Caltech101 | CIFAR100 | DTD | EuroSAT | Flowers | Food | MNIST | OxfordPet | Cars | Sun397 | Avg. |
|---|---|---|---|---|---|---|---|---|---|---|---|---|---|---|
| Zero-shot | ✓ | ✓ | 24.8 | 88.4 | 68.2 | 44.6 | 54.9 | 71.0 | 88.5 | 59.4 | 89.0 | 64.7 | 65.4 | |
| **Transfer** | | | | | | | | | | | | | | |
| LwF (Li & Hoiem, 2017) | ✓ | ✗ | – | 74.5 | 56.9 | 39.1 | 51.1 | 52.6 | 72.8 | 60.6 | 75.1 | 30.3 | 55.9 | 56.9 |
| iCaRL (Rebuffi et al., 2017) | ✓ | ✗ | – | 56.6 | 44.6 | 32.7 | 39.3 | 46.6 | 68.0 | 46.0 | 77.4 | 31.9 | 60.5 | 50.4 |
| LwF-VR (Ding et al., 2022) | ✓ | ✗ | – | 77.1 | 61.0 | 40.5 | 45.3 | 54.4 | 74.6 | 47.9 | 76.7 | 36.3 | 58.6 | 57.2 |
| WiSE-FT (Wortsman et al., 2022) | ✓ | ✗ | – | 73.5 | 55.6 | 35.6 | 41.5 | 47.0 | 68.3 | 53.9 | 69.3 | 26.8 | 51.9 | 52.3 |
| ZSCL (Zheng et al., 2023) | ✓ | ✗ | – | **86.0** | 67.4 | 45.4 | 50.4 | 69.1 | 87.6 | 61.8 | 86.8 | **60.1** | 66.8 | 68.1 |
| O-LoRA (Wang et al., 2023a) | ✓ | ✓ | – | 80.8 | 68.0 | 44.5 | 49.8 | 67.5 | 86.7 | 59.3 | 88.7 | 56.1 | 63.6 | 66.5 |
| InfLoRA (Liang & Li, 2024) | ✓ | ✓ | – | 84.3 | 67.4 | 44.3 | 50.6 | 68.2 | 87.1 | 62.7 | 88.7 | 57.8 | 62.8 | 67.4 |
| SD-LoRA (Wu et al., 2025b) | ✓ | ✓ | – | 82.3 | 67.5 | 44.4 | 51.0 | 67.9 | 87.2 | 61.1 | 88.4 | 58.2 | 63.4 | 67.1 |
| KeepLoRA | ✓ | ✓ | – | 84.6 | 68.7 | 45.9 | 54.3 | 70.1 | 87.7 | 64.8 | 90.3 | 59.5 | 64.1 | **69.0** |
| L2P (Wang et al., 2022c) | ✗ | ✓ | – | 65.6 | 50.9 | 30.4 | 41.4 | 49.3 | 71.8 | 36.3 | 77.5 | 55.3 | 53.4 | 53.2 |
| DualPrompt (Wang et al., 2022b) | ✗ | ✓ | – | 56.7 | 51.4 | 28.7 | 33.7 | 45.6 | 70.9 | 59.5 | 77.7 | 49.5 | 50.4 | 52.4 |
| S-Prompts (Wang et al., 2022a) | ✗ | ✓ | – | 67.3 | 49.4 | 26.7 | 39.7 | 47.1 | 70.2 | 34.3 | 78.9 | 56.7 | 52.2 | 52.2 |
| DIKI (Tang et al., 2024) | ✗ | ✓ | – | 92.9 | 69.1 | 43.2 | 43.9 | 65.4 | 85.3 | 56.0 | 88.4 | 64.0 | 65.6 | 67.4 |
| MoE-Adapters (Yu et al., 2024) | ✗ | ✗ | – | 87.9 | 68.2 | 44.4 | 49.9 | 70.7 | 88.7 | 59.7 | 89.1 | 64.5 | 65.5 | 68.9 |
| IAP (Fu et al., 2025) | ✗ | ✓ | – | **93.0** | 68.7 | 44.0 | 47.0 | 70.4 | 85.9 | 63.5 | 89.7 | 66.2 | 63.3 | 69.2 |
| KeepLoRA+ | ✗ | ✓ | – | 85.9 | 69.9 | 44.6 | 53.7 | 70.9 | 88.9 | 65.4 | 90.8 | 63.0 | 66.1 | **69.9** |
| **Average** | | | | | | | | | | | | | | |
| LwF (Li & Hoiem, 2017) | ✓ | ✗ | 36.3 | 86.9 | 72.0 | 59.0 | 73.7 | 60.0 | 73.6 | 74.8 | 80.0 | 37.3 | 58.1 | 64.7 |
| iCaRL (Rebuffi et al., 2017) | ✓ | ✗ | 35.5 | 89.2 | 72.2 | 60.6 | 68.8 | 70.0 | 78.2 | 62.3 | 81.8 | 41.2 | 62.5 | 65.7 |
| LwF-VR (Ding et al., 2022) | ✓ | ✗ | 29.6 | 87.7 | 74.4 | 59.5 | 72.4 | 63.6 | 77.0 | 66.7 | 81.2 | 43.7 | 60.7 | 65.1 |
| WiSE-FT (Wortsman et al., 2022) | ✓ | ✗ | 26.7 | 86.5 | 64.3 | 57.1 | 65.7 | 58.7 | 71.1 | 70.5 | 75.8 | 36.9 | 54.6 | 60.7 |
| ZSCL (Zheng et al., 2023) | ✓ | ✗ | 45.1 | 92.0 | 80.1 | 64.3 | 79.5 | 81.6 | 89.6 | 75.2 | 88.9 | **64.7** | 68.0 | 75.4 |
| O-LoRA (Wang et al., 2023a) | ✓ | ✓ | 39.8 | 93.2 | 78.3 | 61.7 | 78.9 | 76.3 | 88.5 | 73.9 | 90.1 | 60.2 | 65.2 | 73.3 |
| InfLoRA (Liang & Li, 2024) | ✓ | ✓ | 53.6 | 95.6 | 82.8 | 65.0 | 80.9 | 79.6 | 89.1 | 76.1 | 90.2 | 62.3 | 64.5 | 76.3 |
| SD-LoRA (Wu et al., 2025b) | ✓ | ✓ | 36.7 | 92.2 | 80.2 | 55.9 | 77.5 | 73.2 | 89.2 | 74.9 | 89.8 | 62.5 | 65.0 | 72.5 |
| KeepLoRA | ✓ | ✓ | **55.6** | 95.7 | 83.2 | 65.6 | 82.2 | 82.0 | 89.5 | 77.4 | 91.5 | 63.9 | 65.8 | **77.5** |
| L2P (Wang et al., 2022c) | ✗ | ✓ | 38.0 | 85.2 | 78.2 | 61.3 | 72.9 | 74.9 | 79.7 | 59.1 | 82.0 | 59.7 | 55.4 | 67.9 |
| DualPrompt (Wang et al., 2022b) | ✗ | ✓ | 37.8 | 84.3 | 78.6 | 60.1 | 71.1 | 73.2 | 79.1 | 73.9 | 82.3 | 55.1 | 52.8 | 68.0 |
| S-Prompts (Wang et al., 2022a) | ✗ | ✓ | 37.5 | 92.5 | 77.5 | 58.2 | 76.4 | 74.1 | 78.8 | 57.9 | 83.0 | 60.8 | 54.4 | 68.3 |
| DIKI (Tang et al., 2024) | ✗ | ✓ | 45.4 | 95.7 | 83.0 | 65.0 | 78.2 | 82.5 | 87.1 | 71.7 | 90.0 | 67.2 | 66.6 | 75.7 |
| MoE-Adapters (Yu et al., 2024) | ✗ | ✗ | 50.2 | 91.9 | 83.1 | 69.4 | 78.9 | 84.0 | 89.1 | 73.7 | 89.3 | 67.7 | 66.9 | 76.7 |
| IAP (Fu et al., 2025) | ✗ | ✓ | 45.9 | 95.8 | 83.3 | 66.5 | 79.5 | 84.8 | 87.5 | 76.6 | 91.0 | 69.2 | 64.5 | 76.8 |
| KeepLoRA+ | ✗ | ✓ | **58.4** | 96.5 | 84.4 | 67.8 | 82.1 | 84.5 | 90.7 | 77.8 | 91.9 | 67.5 | 67.6 | **79.0** |
| **Last** | | | | | | | | | | | | | | |
| LwF (Li & Hoiem, 2017) | ✓ | ✗ | 26.3 | 87.5 | 71.9 | 66.6 | 79.9 | 66.9 | 83.8 | 99.6 | 92.1 | 66.1 | 80.4 | 74.6 |
| iCaRL (Rebuffi et al., 2017) | ✓ | ✗ | 35.8 | 93.0 | 77.0 | 70.2 | 83.3 | 88.5 | 90.4 | 86.7 | 93.2 | 81.2 | 81.9 | 80.1 |
| LwF-VR (Ding et al., 2022) | ✓ | ✗ | 20.5 | 89.8 | 72.3 | 67.6 | 85.5 | 73.8 | 85.7 | 99.6 | 93.1 | 73.3 | 80.9 | 76.6 |
| WiSE-FT (Wortsman et al., 2022) | ✓ | ✗ | 27.2 | 90.8 | 68.0 | 68.9 | 86.9 | 74.0 | 87.6 | 99.6 | 92.6 | 77.8 | 81.3 | 77.7 |
| ZSCL (Zheng et al., 2023) | ✓ | ✗ | 40.6 | 92.2 | 81.3 | 70.5 | 94.8 | 90.5 | 91.9 | 98.7 | 93.9 | 85.3 | 80.2 | 83.6 |
| O-LoRA (Wang et al., 2023a) | ✓ | ✓ | 31.4 | 91.8 | 75.7 | 61.1 | 89.0 | 76.0 | 88.9 | 99.1 | 92.3 | 74.8 | 81.3 | 78.3 |
| InfLoRA (Liang & Li, 2024) | ✓ | ✓ | 51.1 | 96.5 | 85.1 | 70.7 | 98.1 | 87.7 | 91.3 | 99.4 | 94.2 | 82.0 | 81.4 | 85.2 |
| SD-LoRA (Wu et al., 2025b) | ✓ | ✓ | 31.1 | 92.3 | 79.8 | 57.4 | 88.7 | 76.1 | 90.6 | 99.0 | 92.9 | 81.3 | 81.6 | 79.2 |
| KeepLoRA | ✓ | ✓ | 53.2 | 96.8 | 85.7 | 71.4 | 98.1 | 90.8 | 91.4 | 99.6 | 94.5 | 83.1 | 82.0 | **86.1** |
| L2P (Wang et al., 2022c) | ✗ | ✓ | 38.0 | 87.1 | 84.2 | 72.9 | 86.0 | 96.1 | 89.2 | 99.0 | 94.1 | 79.6 | 76.0 | 82.0 |
| DualPrompt (Wang et al., 2022b) | ✗ | ✓ | 37.8 | 87.1 | 84.6 | 71.8 | 89.2 | 96.3 | 89.1 | 99.1 | 94.5 | 79.9 | 76.5 | 82.3 |
| S-Prompts (Wang et al., 2022a) | ✗ | ✓ | 37.5 | 95.1 | 83.7 | 70.2 | 97.5 | 96.5 | 89.0 | 99.1 | 94.0 | 79.5 | 75.8 | 83.4 |
| DIKI (Tang et al., 2024) | ✗ | ✓ | 45.4 | 95.9 | 86.0 | 73.0 | 97.8 | 96.8 | 89.3 | 99.3 | 94.4 | 81.8 | 76.4 | 85.1 |
| MoE-Adapters (Yu et al., 2024) | ✗ | ✗ | 49.8 | 92.2 | 86.1 | 78.1 | 95.7 | 94.3 | 89.5 | 98.1 | 89.9 | 81.6 | 80.0 | 85.0 |
| IAP (Fu et al., 2025) | ✗ | ✓ | 46.8 | 96.1 | 86.7 | 75.2 | 98.1 | 97.0 | 89.6 | 99.4 | 94.7 | 82.8 | 76.7 | 85.7 |
| KeepLoRA+ | ✗ | ✓ | **57.3** | 97.6 | 87.2 | 76.5 | 98.4 | 95.7 | 92.6 | 99.5 | 94.7 | 87.2 | 83.2 | **88.2** |

that the initialization technique of $A_t$ in KeepLoRA satisfies these properties, which ensures that span($A_t$) is orthogonal to the principal subspace of the model parameters $W_p$ and dominant feature directions of each learned task $M_{t-1}$ to maintain stability, while being adaptive to the current task gradient to improve plasticity.

# 4 EXPERIMENTS

We conduct experiments on various benchmarks to validate the effectiveness of KeepLoRA in balancing three core objectives of continual learning: forward stability, backward stability, and plasticity. (i) To quantify forward forgetting, we calculate the average accuracy on tasks $t + 1, \ldots, n$ after training on task $t$, which is defined as the *Transfer* metric, presented in Tab. 2, 3 and 4. Fig. 3 further analyzes how KeepLoRA maintains the transferability. (ii) The *Last* metric, shown in Tab. 2, 3 and 4, assesses model performance after continual training has completed, capturing both plasticity and backward stability. (iii) To further analyze plasticity, Fig. 2 compares our method with an unconstrained LoRA, demonstrating that KeepLoRA preserves stability with minimal sacrifice to its adaptive capability. The *Average* metric represents the mean accuracy across all learned tasks, offering a holistic measure of the balance between stability and plasticity.

Table 3: Comparison of different continual learning methods on MLLM-DCL benchmark for **VQA** tasks in terms of *Transfer*, *Average*, and *Last* scores (%). The best results are in **bold**.

| Method | Sensing | Medical | Driving | Science | Finance | Avg. |
|---|---|---|---|---|---|---|
| Zero-shot | 32.29 | 28.28 | 15.59 | 35.55 | 62.56 | |
| **Transfer** | | | | | | |
| LoRA-FT (Hu et al., 2022) | – | 28.10 | 17.44 | 34.03 | 50.19 | 32.44 |
| O-LoRA (Wang et al., 2023a) | – | 28.37 | 18.37 | 33.72 | 52.53 | 33.25 |
| CL-MoE (Huai et al., 2025) | – | 28.25 | 19.38 | 34.08 | 48.56 | 32.57 |
| SEFE (Chen et al., 2025) | – | 28.10 | **19.63** | 33.85 | 52.36 | 33.49 |
| KeepLoRA | – | **28.49** | 16.63 | **34.13** | **55.61** | **33.71** |
| **Average** | | | | | | |
| LoRA-FT (Hu et al., 2022) | 73.34 | 44.94 | 31.38 | 38.79 | 57.84 | 49.26 |
| O-LoRA (Wang et al., 2023a) | 75.04 | 45.71 | 32.62 | 38.54 | 59.64 | 50.31 |
| CL-MoE (Huai et al., 2025) | 74.19 | 45.60 | 32.08 | 38.88 | 56.68 | 49.49 |
| SEFE (Chen et al., 2025) | 77.71 | 47.69 | 35.35 | 38.99 | 59.57 | 51.86 |
| KeepLoRA | **79.55** | **50.80** | **37.53** | **40.70** | **62.35** | **54.19** |
| **Last** | | | | | | |
| LoRA-FT (Hu et al., 2022) | 69.34 | 44.30 | 29.10 | 41.44 | 88.43 | 54.52 |
| O-LoRA (Wang et al., 2023a) | 72.30 | 46.89 | 31.59 | 41.50 | 88.06 | 56.07 |
| CL-MoE (Huai et al., 2025) | 71.83 | 47.36 | 29.49 | 41.48 | 89.16 | 55.86 |
| SEFE (Chen et al., 2025) | 77.05 | 50.86 | 40.27 | 42.98 | 88.40 | 59.91 |
| KeepLoRA | **78.76** | **54.34** | **50.19** | **49.48** | **89.30** | **64.41** |

## 4.1 MAIN RESULTS

We evaluate our method on the dual-encoder model CLIP (Radford et al., 2021) and encoder-decoder model LLaVA (Liu et al., 2023). For CLIP, the experiments are conducted on the MTIL (Zheng et al., 2023) benchmark, presenting results for alphabetical (Tab. 2) and random (Tab. 6) task orders in two settings, with and without architecture extension. KeepLoRA+ is a structure extension variant with a prototype vector for a class name to help classification, detailed in Appendix B.3. For LLaVA, the experiments (Tab. 3 and 4) are conducted on MLLM-DCL (Guo et al., 2025b) and UCIT (Guo et al., 2025b) benchmarks, including various instruction formats such as image captioning, visual question-answer, and multiple-choice questions. Detailed information on experiment settings and benchmarks is presented in Appendices B.4 and B.1, separately. KeepLoRA and KeepLoRA+ achieve state-of-the-art performance on the *Transfer*, *Average*, and *Last* metrics in each of these settings. This demonstrates that our approach consistently addresses the challenges of forward stability, backward stability, and plasticity in continual learning.

Table 4: Comparison of different continual learning methods on UCIT benchmark for **VQA** tasks in terms of *Transfer*, *Average*, and *Last* scores (%). The best results are in **bold**.

| Method | ImgNet-R | ArxivQA | VizWiz | IconQA | CLEVR | Flickr-30k | Avg. |
|---|---|---|---|---|---|---|---|
| Zero-shot | 16.27 | 53.73 | 38.39 | 19.20 | 20.63 | 41.88 | |
| **Transfer** | | | | | | | |
| LoRA-FT (Hu et al., 2022) | – | 52.63 | 18.30 | 6.02 | 16.97 | 40.29 | 26.84 |
| O-LoRA (Wang et al., 2023a) | – | 52.87 | 19.57 | 4.42 | 16.85 | 41.04 | 26.95 |
| CL-MoE (Huai et al., 2025) | – | 52.00 | 19.32 | 7.37 | 17.81 | 41.28 | 27.56 |
| SEFE (Chen et al., 2025) | – | **53.33** | 18.68 | 7.48 | 17.03 | 40.90 | 27.48 |
| KeepLoRA | – | 52.83 | **20.39** | **9.18** | **18.12** | **41.50** | **28.40** |
| **Average** | | | | | | | |
| LoRA-FT (Hu et al., 2022) | 75.98 | 77.78 | 41.56 | 38.83 | 34.56 | 43.25 | 51.99 |
| O-LoRA (Wang et al., 2023a) | 82.43 | 80.06 | 41.73 | 35.87 | 33.94 | 43.74 | 52.96 |
| CL-MoE (Huai et al., 2025) | 80.16 | 77.10 | 40.43 | 30.33 | 33.10 | 43.95 | 50.85 |
| SEFE (Chen et al., 2025) | 85.49 | 78.55 | **42.92** | **40.33** | 34.80 | 43.64 | 54.29 |
| KeepLoRA | **86.50** | **83.63** | 42.66 | 40.08 | **35.24** | **44.11** | **55.37** |
| **Last** | | | | | | | |
| LoRA-FT (Hu et al., 2022) | 58.60 | 76.73 | 45.72 | 67.43 | 61.57 | **58.03** | 61.35 |
| O-LoRA (Wang et al., 2023a) | 74.17 | 80.93 | 45.30 | 62.87 | 63.83 | 57.24 | 64.06 |
| CL-MoE (Huai et al., 2025) | 67.17 | 75.77 | 44.38 | 52.63 | 54.40 | 57.28 | 58.61 |
| SEFE (Chen et al., 2025) | 80.23 | 79.13 | **47.11** | **69.40** | 65.70 | 57.33 | 66.48 |
| KeepLoRA | **82.43** | **86.70** | 46.54 | 67.80 | **66.40** | 57.18 | **67.84** |

## 4.2 ANALYSIS OF MODEL PLASTICITY

Plasticity assesses the ability to effectively acquire new knowledge following a sequence of continual learning tasks. We evaluate two performance metrics for each task: (i) the accuracy achieved by training on the task in isolation, serving as an upper bound, and (ii) the accuracy measured immediately after the task is learned within the continual sequence. Our analysis in Fig. 2 compares KeepLoRA with a standard LoRA baseline. In the isolation-task setting, KeepLoRA performs comparably to LoRA, as gradient-informed initialization of the frozen down-projection matrix $A$ effectively captures the essential learning direction, maintaining high learning capacity. Furthermore, when switching to the continual learning scenario, KeepLoRA exhibits a consistently

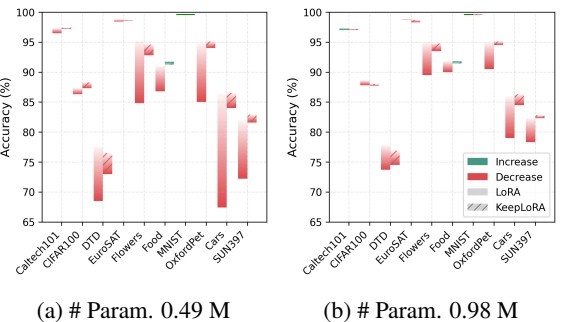

(a) # Param. 0.49 M    (b) # Param. 0.98 M

Figure 2: Comparison of plasticity between KeepLoRA and the LoRA baseline under the same learnable parameter budgets: Fig. 2a 0.49 million parameters and Fig. 2b 0.98 million parameters. Each bar represents the performance drop for a task, measured as the difference between accuracy from isolated training and accuracy after sequential learning and immediate testing.

smaller performance drop on new tasks compared to LoRA. This suggests that, by confining updates to the residual subspace and avoiding interference with previously learned knowledge, our method enhances the model's plasticity for subsequent tasks.

## 4.3 ANALYSIS OF MODEL STABILITY

We analyze stability by visualizing the interference of the LoRA module between multiple tasks in Fig. 3. In these heatmaps, the off-diagonal cells represent inter-task interference, while the vertical bar on the left indicates the overall impact on the backbone. The standard LoRA (Fig. 3a) and LoRA with a frozen matrix $A$ (Fig. 3b) both exhibit significant interference. The bright patterns in their heatmaps and vertical bars show that training on a current task heavily interferes with the

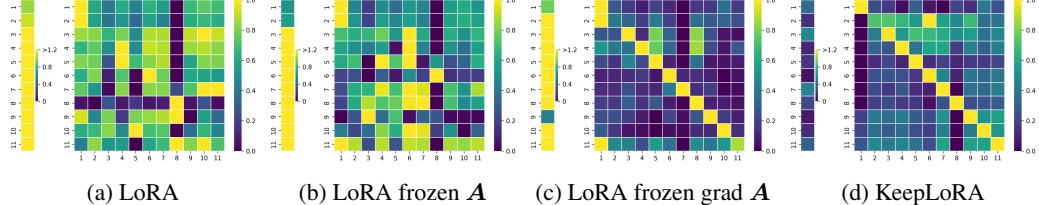

|           |                  |                    |              |
|-----------|------------------|--------------------|--------------|
| (a) LoRA  | (b) LoRA frozen $\boldsymbol{A}$ | (c) LoRA frozen grad $\boldsymbol{A}$ | (d) KeepLoRA |

Figure 3: Visualization of the average L2 norm of the output magnitude from the learned LoRA across multiple tasks. Each heatmap cell at row $i$ and column $j$ displays the normalized average L2 norm of the LoRA's output when the model, trained up to task $i$, is tested on task $j$'s data. The vertical bar to the left of each heatmap indicates the mean output norm across all test tasks after each training stage, with darker colors signifying a lower norm and thus a reduced impact on the stability.

representations of other tasks, leading to poor stability. Although gradient-informed initialization (Fig. 3c) reduces off-diagonal interference, the overall impact on the backbone remains high, as shown by its bright vertical bar. In contrast, KeepLoRA (Fig. 3d) shows a desirable pattern: a bright diagonal with dark off-diagonal cells. This indicates that the updates of the model focus on the current task, causing minimal interference with others. The dark vertical bar further confirms that the overall impact on the backbone is consistently low. By minimizing interference with previously learned tasks, KeepLoRA ensures backward stability. Furthermore, its minimal interference with unseen tasks, as indicated by the low norm, is critical to preserve forward stability.

## 4.4 Ablation Study

To analyze the contribution of each component, we conduct an ablation study starting from a standard LoRA baseline. As shown in Tab. 5, our modifications progressively improve performance. Freezing the down-projection matrix $\boldsymbol{A}$ even with random initialization, enhances stability in the continual learning setting by mitigating destructive interference with the backbone weights. Subsequently, employing a (i) gradient-informed initialization further improves plasticity, leading to a $5.9\%$ increase on the *Last* metric and indicating a more effective adaptation. After constraining the updates to be orthogonal to (ii) the principal subspace $\boldsymbol{W}_p$ and (iii) the dominant feature directions $\boldsymbol{M}$, gains of $4.0\%$ on *Transfer*, $7.3\%$ on *Average*, and $10.7\%$ on *Last*, demonstrating the critical role of subspace projection in balancing stability and plasticity.

Table 5: Ablation Study of KeepLoRA on MTIL.

| Training Strategy | Transfer | $\Delta$ | Average | $\Delta$ | Last | $\Delta$ |
|---|---|---|---|---|---|---|
| LoRA (rank 8, # param. 0.98 M) | 58.3 | 0.0 | 61.5 | 0.0 | 59.4 | 0.0 |
| LoRA frozen $\boldsymbol{A}$ (rank 16, # param. 0.98 M) | 63.9 | +5.6 | 68.2 | +6.7 | 69.5 | +10.1 |
| (i) Replace Eq. 5 by $\hat{\boldsymbol{G}}_t = \boldsymbol{G}_t$ | 65.0 | +6.7 | 70.2 | +8.7 | 75.4 | +16.0 |
| (ii) Replace Eq. 5 by $\hat{\boldsymbol{G}}_t = \boldsymbol{G}_t - \boldsymbol{W}_p \boldsymbol{W}_p^\top \boldsymbol{G}_t$ | 65.9 | +7.6 | 71.5 | +10.0 | 76.5 | +17.1 |
| (iii) Replace Eq. 5 by $\hat{\boldsymbol{G}}_t = \boldsymbol{G}_t - \boldsymbol{M}_{t-1} \boldsymbol{M}_{t-1}^\top \boldsymbol{G}_t$ | 68.1 | +9.8 | 77.2 | +15.7 | 86.1 | +26.7 |
| KeepLoRA (Eq. 5) | 69.0 | +10.7 | 77.5 | +16.0 | 86.1 | +26.7 |

## 5 Conclusion

In this work, we reveal that the principal subspace of parameters encodes general knowledge and the residual subspace captures domain-specific adaptations. Leveraging this insight, we propose KeepLoRA, a parameter-efficient fine-tuning method that can effectively achieve a balance among the competing objectives of plasticity, backward stability, and forward stability. Our theoretical analysis confirms that constraining parameter updates to the residual subspace is an optimal strategy, maximizing plasticity for the current task while maintaining orthogonality to subspaces encoding general and previously learned knowledge. Empirically, KeepLoRA matches the learning capacity of unconstrained LoRA on isolated tasks while significantly mitigating forgetting in continual learning scenarios. As a simple and effective method, KeepLoRA provides a principled approach for continual learning that is applicable to larger models and more diverse tasks.

## ACKNOWLEDGEMENTS

This work was supported by the National Science Foundation of China (62576092, 62225602), and the Big Data Computing Center of Southeast University. We would like to thank anonymous reviewers for their constructive suggestions.

## ETHICS STATEMENT

All authors have read and adhered to the ICLR Code of Ethics. This paper presents an algorithmic contribution, KeepLoRA, aimed at advancing the field of continual learning. Our empirical validation is conducted exclusively on publicly available and widely used academic benchmarks, such as CIFAR100 and Caltech101, which do not contain personally identifiable or sensitive information. While we acknowledge that advancements in machine learning have broad societal consequences, our work does not introduce foreseeable negative applications or exacerbate biases beyond those potentially present in the general pre-trained models.

## REPRODUCIBILITY STATEMENT

To ensure full reproducibility, we have open-sourced the code for our method. Also, the method is detailed in Sec. 3, with the core framework summarized in Algorithm 1. We specify all hyperparameters used for our method and the baselines, including learning rates, batch size, and the preservation ratios $\epsilon_w$ and $\epsilon_f$ in Appendix B.

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

# A PROOFS OF PROPOSITIONS

## A.1 PROOFS OF PROPOSITION 3.1

*Proof.* Suppose for loss function $\mathcal{L}$ for task $\mathcal{T}^t$ and a linear layer with $y = xW$, where $y$ is the output of the layer and $x$ is the input. We can compute the gradient of $B_t$ directly as follows:

$$\frac{\partial \mathcal{L}}{\partial B_t} = \frac{\partial W}{\partial B_t} \cdot \frac{\partial \mathcal{L}}{\partial W} = \frac{\alpha}{r} A_t^\top G_t. \tag{9}$$

In a gradient descent iteration, the change of $B_t$ is represented by a negative gradient: $\Delta B_t = -\frac{\eta\alpha}{r} A_t^\top G_t$. Therefore, when $A_t$ is frozen to only update $B_t$ in each iteration, we can obtain the variation of $W$ in one iteration to complete the proof:

$$\Delta W = \frac{\alpha}{r} A_t \Delta B_t = -\frac{\eta\alpha^2}{r^2} A_t A_t^\top G_t. \tag{10}$$

$\square$

## A.2 PROOFS OF PROPOSITION 3.2

*Proof.* We proceed by transforming the constrained optimization problem, leveraging subspace properties, and applying the Eckart–Young–Mirsky Theorem (Eckart & Young, 1936) to confirm the optimal solution.

*Step 1: Equivalent Transformation of the Objective Function.* For an orthonormal matrix $A_t$ satisfying $A_t^\top A_t = I$, the orthogonal projection operator $P_{A_t} = A_t A_t^\top$ satisfies the Pythagorean theorem for the Frobenius norm:

$$\|G_t\|_F^2 = \|P_{A_t} G_t\|_F^2 + \|G_t - P_{A_t} G_t\|_F^2.$$

Since $\|G_t\|_F^2$ is a constant independent of $A_t$, minimizing the original objective $\|G_t - P_{A_t} G_t\|_F^2$ is *equivalent* to maximizing the projected norm $\|P_{A_t} G_t\|_F^2$. The optimization problem thus can be rewritten as:

$$\max_{A_t^\top A_t = I} \|A_t A_t^\top G_t\|_F^2,$$
$$\text{s.t} \quad W_p^\top A_t = M_{t-1}^\top A_t = 0. \tag{11}$$

*Step 2: Substitute $\hat{G}_t$ and Simplify Using Constraints.* Recall from Eq. 5 that the projected gradient $\hat{G}_t$ is defined as:

$$\hat{G}_t = G_t - W_p W_p^\top G_t - M_{t-1} M_{t-1}^\top G_t.$$

Rearranging gives $G_t = \hat{G}_t + W_p W_p^\top G_t + M_{t-1} M_{t-1}^\top G_t$. Substitute this into the objective:

$$\|A_t A_t^\top \left( \hat{G}_t + W_p W_p^\top G_t + M_{t-1} M_{t-1}^\top G_t \right)\|_F^2.$$

For any feasible $A_t$, we use $W_p^\top A_t = M_{t-1}^\top A_t = 0$ to simplify: $A_t^\top (W_p W_p^\top G_t) = (W_p^\top A_t)^\top (W_p^\top G_t) = 0^\top (W_p^\top G_t) = 0$. Similarly, $A_t^\top (M_{t-1} M_{t-1}^\top G_t) = 0$.

Thus, $A_t A_t^\top (W_p W_p^\top G_t + M_{t-1} M_{t-1}^\top G_t) = 0$, and the objective reduces to maximizing $\|A_t A_t^\top \hat{G}_t\|_F^2$. The optimization problem simplifies to:

$$\max_{A_t^\top A_t = I} \|A_t A_t^\top \hat{G}_t\|_F^2,$$
$$\text{s.t} \quad W_p^\top A_t = M_{t-1}^\top A_t = 0. \tag{12}$$

*Step 3: Optimal $A_t$ via Eckart–Young–Mirsky Theorem.* The Eckart–Young–Mirsky Theorem (Eckart & Young, 1936) states that for any matrix $X \in \mathbb{R}^{m \times n}$ and integer $k \leq \min(m, n)$, the $r$-dimensional subspace that maximizes $\|PX\|_F^2$, where $P$ is the orthogonal projection onto the subspace, is spanned by the top-$r$ left singular vectors of $X$.

Here, $\boldsymbol{X} = \hat{\boldsymbol{G}}_t$, and we seek an $r$-dimensional subspace spanned by $\boldsymbol{A}_t$ to maximize $\|\boldsymbol{A}_t \boldsymbol{A}_t^\top \hat{\boldsymbol{G}}_t\|_F^2$. By the theorem, the optimal $\boldsymbol{A}_t$ consists of the top-$r$ left singular vectors of $\hat{\boldsymbol{G}}_t$.

*Step 4: Verify Feasibility of the Optimal $\boldsymbol{A}_t$.* We confirm the optimal $\boldsymbol{A}_t$ satisfies the constraints $\boldsymbol{W}_p^\top \boldsymbol{A}_t = \boldsymbol{0}$ and $\boldsymbol{M}_{t-1}^\top \boldsymbol{A}_t = \boldsymbol{0}$.

By the definition of $\hat{\boldsymbol{G}}_t$ in Eq. 5, we have:

$$\boldsymbol{W}_p^\top \hat{\boldsymbol{G}}_t = \boldsymbol{0}, \quad \boldsymbol{M}_{t-1}^\top \hat{\boldsymbol{G}}_t = \boldsymbol{0}. \tag{13}$$

Substituting SVD of $\hat{\boldsymbol{G}} = \boldsymbol{U}\boldsymbol{S}\boldsymbol{V}$ in Eq. 13 : $\boldsymbol{W}_p^\top \hat{\boldsymbol{G}}_t = \boldsymbol{W}_p^\top \boldsymbol{U}\boldsymbol{S}\boldsymbol{V}^\top = \boldsymbol{0}$. Since $\boldsymbol{S}\boldsymbol{V}^\top$ is column-full rank (singular values are non-negative, and $\boldsymbol{V}$ is orthonormal), $\boldsymbol{W}_p^\top \boldsymbol{U}$ must be the zero matrix. Thus, $\boldsymbol{W}_p^\top \boldsymbol{U} = \boldsymbol{0}$, hence $\boldsymbol{W}_p^\top \boldsymbol{A}_t = \boldsymbol{W}_p^\top \boldsymbol{U}_{:,1:r} = \boldsymbol{0}$. The same logic applies to $\boldsymbol{M}_{t-1}$: $\boldsymbol{M}_{t-1}^\top \hat{\boldsymbol{G}}_t = \boldsymbol{M}_{t-1}^\top \boldsymbol{U}\boldsymbol{S}\boldsymbol{V}^\top = \boldsymbol{0}$ implies $\boldsymbol{M}_{t-1}^\top \boldsymbol{U} = \boldsymbol{0}$, hence $\boldsymbol{M}_{t-1}^\top \boldsymbol{A}_t = \boldsymbol{0}$.

Thus, the optimal solution to Eq. 8 is exactly the top-$r$ left singular vectors of $\hat{\boldsymbol{G}}_t$, which matches KeepLoRA $\boldsymbol{A}_t$ initialization. The proof is completed. $\square$

# B  EXPERIMENT DETAILS

## B.1  BENCHMARK

**MTIL** benchmark (Zheng et al., 2023) consists of 11 image classification datasets: Aircraft (Maji et al., 2013), Caltech101 (Fei-Fei et al., 2004), Cifar100 (Krizhevsky et al., 2009), DTD (Cimpoi et al., 2014), EuroSAT (Helber et al., 2019), Flowers (Nilsback & Zisserman, 2008), Food (Bossard et al., 2014), MNIST (Deng, 2012), OxfordPet (Parkhi et al., 2012), StanfordCars (Krause et al., 2013), and SUN397 (Xiao et al., 2010). Each dataset is treated as a task.

**MLLM-DCL** benchmark (Zhao et al., 2025) consists of multiple downstream VQA datasets: RSVQA (Lobry et al., 2020), PathVQA (He et al., 2020), DriveLM (Sima et al., 2024), FinVis (Wang et al., 2023b), AI2D (Kembhavi et al., 2016), Sciverse (Guo et al., 2025c), MapQA (Chang et al., 2022), and TQA (Kembhavi et al., 2017). It covers 5 specialized areas: Remote Sensing, Medical, Driving, Finance, and Science. Each area is treated as a task.

**UCIT** benchmark (Guo et al., 2025a) consists of 6 VQA datasets: ArxivQA (Li et al., 2024), CLEVR-Math (Lindström & Abraham, 2022), IconQA (Lu et al., 2021), ImageNet-R (Hendrycks et al., 2021), VizWiz-Caption (Gurari et al., 2018), and Flickr30k (Plummer et al., 2015). Each dataset is treated as a task.

## B.2  EVALUATION METRICS

We define the *Transfer*, *Average*, and *Last* metrics to evaluate model performance under continual learning scenarios. Let $a_t^{(i)}$ represent the accuracy of the model on task $t$ after training on task $i$ with a total of $n$ tasks. The *Transfer*, *Average*, and *Last* metrics for task $t$ are computed as follows:

$$\text{Transfer}_t = \frac{1}{t-1} \sum_{i=1}^{t-1} a_t^{(i)}, \quad t = 2, 3, \ldots, n, \tag{14}$$

$$\text{Average}_t = \frac{1}{n} \sum_{i=1}^{n} a_t^{(i)}, \quad t = 1, 2, \ldots, n, \tag{15}$$

$$\text{Last}_t = a_t^{(n)}, \quad t = 1, 2, \ldots, n. \tag{16}$$

The *Transfer* metric evaluates forward stability by measuring the performance of unseen tasks throughout $(i+1, i+2, \ldots, n)$ after training on the task $i$. The *Last* metric measures the final performance on each task after completing all training steps, quantifying both plasticity and backward stability. The *Average* metric represents the mean accuracy across all time steps, offering a holistic measure of stability and plasticity.

### B.3 Implementation Details of KeepLoRA+

We extend KeepLoRA with a structure variant, termed KeepLoRA+, which incorporates a prototype vector for each class name to improve classification performance. Each prototype vector is initialized using the mean feature extracted by the vision encoder from the corresponding class samples. During the training stage, we jointly optimize the prototype vectors alongside the KeepLoRA parameters. In the inference stage, the logits derived from the similarity of the prototype vectors are averaged with the logits calculated from the text-side contrast.

### B.4 Additional Implementation Details

**CLIP Experiments.** We adopt the CLIP (Radford et al., 2021) model with a ViT-B/16 (Dosovitskiy et al., 2021) image encoder. The training process is carried out using the AdamW (Loshchilov & Hutter, 2019) optimizer, with a learning rate of $10^{-3}$ and a batch size of $64$ across all tasks with no more than $10$ epochs. For the primary experiments, we set the hyperparameters as $\epsilon_{w(\text{vision})} = 0.85$ and $\epsilon_{w(\text{text})} = 0.2$ in vision encoder and text encoder separately and set $\epsilon_f = 0.99$. KeepLoRA+ is a structure extension variant with an extension prototype vector for a classname to help classification. All experiments of KeepLoRA are conducted on a single NVIDIA 4090 GPU. For the reproduced methods, we performed careful hyperparameter tuning. For O-LoRA (Wang et al., 2023a), the learning rate is $5 \times 10^{-4}$ with a regularization coefficient of $0.1$. For InfLoRA (Liang & Li, 2024), the learning rate is $10^{-3}$, with $\epsilon_f = 0.99$. The learning rate for SD-LoRA (Wu et al., 2025b) is set to $5 \times 10^{-3}$.

**LLaVA Experiments.** We adopt the LLaVA-1.5-7b (Liu et al., 2023) model for multimodal continual instruction tuning experiments. The training is conducted on $4\times$ NVIDIA H100 GPUs using the AdamW optimizer. For the MLLM-DCL benchmark, we set the learning rate to $2 \times 10^{-5}$ and train for no more than $3$ epochs per task. For the UCIT benchmark, the learning rate is set to $2 \times 10^{-4}$ for all tasks except Flickr30k, which uses $1 \times 10^{-4}$ and train $1$ epoch for each task. The hyperparameters for subspace constraints are configured as $\epsilon_w = 0.6$ and $\epsilon_f = 0.99$.

## C Supplementary Experiments

### C.1 Comparison on MTIL with order II.

We compare different methods on MTIL in random order: StanfordCars, Food, MNIST, OxfordPet, Flowers, SUN397, Aircraft, Caltech101, DTD, EuroSAT and CIFAR100. As shown in Tab. 6, KeepLoRA consistently outperforms previous methods across all metrics.

### C.2 Hyperparameter Analysis

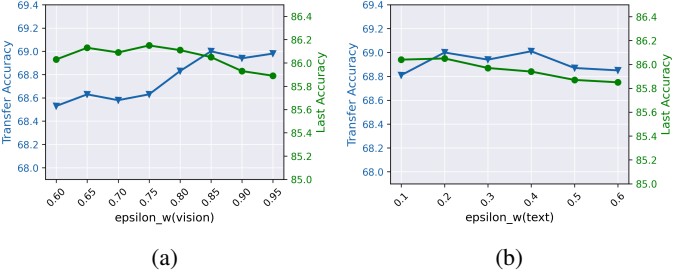

(a)          (b)

Figure 4: Effects of hyperparameters $\epsilon_{w(\text{vision})}$ and $\epsilon_{w(\text{text})}$ on *Transfer* and *Last*, respectively.

We examine the effects of hyperparameters $\epsilon_{w(\text{vision})}$ and $\epsilon_{w(\text{text})}$ on *Transfer* and *Last*. For the image encoder, *Last* fluctuates slightly, with a minor decline when $\epsilon_{w(\text{vision})}$ is larger. Between 0.75 and 0.85, *Transfer* shows a clear increase. For the text encoder, which uses only class names and thus has much less training data than images, the coefficient $\epsilon_{w(\text{text})}$ exhibits low performance sensitivity. Datasets with image-text pairs or VQA tasks, which include substantial text data, warrant further study in this regard.

Table 6: Comparison of different continual learning methods on MTIL for each task with order-II in terms of *Transfer*, *Average*, and *Last* scores (%). The best results are highlighted with **bold** style.

| Method | Inf. Efficiency | w/o Extra Data | Cars | Food | MNIST | OxfordPet | Flowers | Sun397 | Aircraft | Caltech101 | DTD | EuroSAT | CIFAR100 | Avg. |
|---|---|---|---|---|---|---|---|---|---|---|---|---|---|---|
| Zero-shot | ✓ | ✓ | 64.7 | 88.5 | 59.4 | 89.0 | 71.0 | 65.4 | 24.8 | 88.4 | 44.6 | 54.9 | 68.2 | |
| **Transfer** | | | | | | | | | | | | | | |
| LwF (Li & Hoiem, 2017) | ✓ | ✗ | – | 87.8 | **58.5** | 71.9 | 46.6 | 57.3 | 12.8 | 81.4 | 34.5 | 34.5 | 46.8 | 53.2 |
| iCaRL (Rebuffi et al., 2017) | ✓ | ✗ | – | 86.1 | 51.8 | 67.6 | 50.4 | 57.9 | 11.0 | 72.3 | 31.2 | 32.7 | 48.1 | 50.9 |
| LwF-VR (Ding et al., 2022) | ✓ | ✗ | – | 88.2 | 57.0 | 71.4 | 50.0 | 58.0 | 13.0 | 82.0 | 34.4 | 29.3 | 47.6 | 53.1 |
| WiSE-FT (Wortsman et al., 2022) | ✓ | ✗ | – | 87.2 | 57.6 | 67.0 | 45.0 | 54.0 | 12.9 | 78.6 | 35.5 | 28.4 | 44.3 | 51.1 |
| ZSCL (Zheng et al., 2023) | ✓ | ✗ | – | 88.3 | 57.5 | 84.7 | 68.1 | 64.8 | 21.1 | 88.2 | 45.3 | **55.2** | 68.2 | 64.1 |
| O-LoRA (Wang et al., 2023a) | ✓ | ✓ | – | 87.8 | 56.7 | 90.1 | 71.4 | 64.0 | 20.7 | 87.4 | 43.9 | 46.3 | 65.9 | 63.4 |
| InfLoRA (Liang & Li, 2024) | ✓ | ✓ | – | 88.2 | 56.7 | 90.2 | 71.3 | 65.0 | 22.2 | 88.2 | 43.8 | 47.3 | 67.2 | 64.0 |
| SD-LoRA (Wu et al., 2025b) | ✓ | ✓ | – | 88.0 | 56.4 | 90.5 | 71.0 | 64.6 | 22.0 | 87.8 | 43.7 | 47.1 | 66.4 | 63.7 |
| KeepLoRA | ✓ | ✓ | – | **88.7** | 57.7 | 91.2 | 72.1 | 65.8 | 23.4 | 88.8 | 45.4 | 48.5 | 68.2 | **65.0** |
| L2P (Wang et al., 2022c) | ✗ | ✓ | – | 70.6 | 30.7 | 78.3 | 42.8 | 38.3 | 17.4 | 75.3 | 27.4 | 23.1 | 20.7 | 42.5 |
| DualPrompt (Wang et al., 2022b) | ✗ | ✓ | – | 79.9 | 46.9 | 85.2 | 51.3 | 45.1 | 9.3 | 82.7 | 29.9 | 42.9 | 47.2 | 52.1 |
| S-Prompts (Wang et al., 2022a) | ✗ | ✓ | – | 59.8 | 46.2 | 67.7 | 47.5 | 43.8 | 13.5 | 76.8 | 31.4 | 22.6 | 43.5 | 45.3 |
| DIKI (Tang et al., 2024) | ✗ | ✓ | – | 85.8 | **59.8** | 89.1 | 71.8 | 62.6 | 24.3 | 93.3 | 42.7 | 46.8 | 67.8 | 64.4 |
| MoE-Adapters (Yu et al., 2024) | ✗ | ✗ | – | 88.8 | 59.5 | 89.1 | 69.9 | 64.4 | 18.1 | 86.9 | 43.7 | 54.6 | 68.2 | 64.3 |
| IAP (Fu et al., 2025) | ✗ | ✓ | – | 85.7 | 59.4 | 89.1 | 71.3 | 62.7 | 24.4 | 94.0 | 43.8 | 49.0 | 68.6 | 64.9 |
| KeepLoRA+ | ✗ | ✓ | – | **89.1** | 58.1 | 90.7 | 72.4 | 65.4 | 24.0 | 88.9 | 44.0 | 52.7 | 70.2 | **65.5** |
| **Average** | | | | | | | | | | | | | | |
| LwF (Li & Hoiem, 2017) | ✓ | ✗ | 49.0 | 77.0 | 92.1 | 85.9 | 66.5 | 67.2 | 20.9 | 84.7 | 44.6 | 45.5 | 50.5 | 62.2 |
| iCaRL (Rebuffi et al., 2017) | ✓ | ✗ | 52.0 | 75.9 | 77.4 | 74.6 | 58.4 | 59.3 | 11.7 | 79.6 | 42.1 | 43.2 | 51.7 | 56.9 |
| LwF-VR (Ding et al., 2022) | ✓ | ✗ | 44.9 | 75.8 | 91.8 | 85.3 | 63.5 | 67.6 | 16.9 | 84.9 | 44.0 | 40.6 | 51.3 | 60.6 |
| WiSE-FT (Wortsman et al., 2022) | ✓ | ✗ | 52.6 | 79.3 | 91.9 | 83.9 | 63.4 | 65.2 | 23.3 | 83.7 | 45.4 | 40.0 | 48.2 | 61.5 |
| ZSCL (Zheng et al., 2023) | ✓ | ✗ | 81.7 | 91.3 | 91.9 | 91.0 | 82.9 | 72.5 | 33.6 | 89.7 | **53.3** | 62.8 | 69.9 | 74.6 |
| O-LoRA (Wang et al., 2023a) | ✓ | ✓ | 78.5 | 91.0 | 91.3 | 92.3 | 77.7 | 73.0 | 33.5 | 90.5 | 50.7 | 55.1 | 67.8 | 72.9 |
| InfLoRA (Liang & Li, 2024) | ✓ | ✓ | 84.0 | 92.1 | 91.7 | 93.2 | 81.6 | 74.3 | 34.3 | 91.3 | 51.5 | 56.6 | 69.0 | 74.5 |
| SD-LoRA (Wu et al., 2025b) | ✓ | ✓ | 76.8 | 91.1 | 90.8 | 92.5 | 76.5 | 73.1 | 34.0 | 90.7 | 49.1 | 56.2 | 68.2 | 72.6 |
| KeepLoRA | ✓ | ✓ | **85.2** | 92.3 | 92.0 | 93.7 | 84.8 | 74.8 | 35.9 | 91.8 | 53.1 | 57.5 | 70.0 | **75.6** |
| L2P (Wang et al., 2022c) | ✗ | ✓ | 80.1 | 87.4 | 86.7 | 89.6 | 76.8 | 59.1 | 27.7 | 79.5 | 39.9 | 34.6 | 26.5 | 62.5 |
| DualPrompt (Wang et al., 2022b) | ✗ | ✓ | 78.6 | 88.4 | 89.7 | 91.7 | 80.0 | 62.4 | 23.2 | 85.0 | 41.3 | 51.6 | 50.7 | 67.5 |
| S-Prompts (Wang et al., 2022a) | ✗ | ✓ | 79.2 | 86.5 | 89.5 | 87.0 | 78.2 | 61.5 | 25.5 | 83.6 | 41.9 | 36.3 | 47.2 | 65.1 |
| DIKI (Tang et al., 2024) | ✗ | ✓ | 81.9 | 88.9 | 92.1 | 92.8 | 87.7 | 70.3 | 34.3 | 94.2 | 51.5 | 56.1 | 69.5 | 74.5 |
| MoE-Adapters (Yu et al., 2024) | ✗ | ✗ | 84.9 | 89.9 | 89.3 | 91.4 | 86.2 | 72.2 | 33.4 | 89.4 | **53.3** | 61.4 | 69.9 | 74.7 |
| IAP (Fu et al., 2025) | ✗ | ✓ | 82.5 | 89.2 | 92.3 | 93.2 | 88.0 | 70.4 | 34.3 | 94.4 | 52.4 | 57.9 | 70.2 | 75.1 |
| KeepLoRA+ | ✗ | ✓ | **88.0** | 92.4 | 91.9 | 93.9 | 87.4 | 75.2 | 39.2 | 92.0 | 52.8 | 60.9 | 71.8 | **76.9** |
| **Last** | | | | | | | | | | | | | | |
| LwF (Li & Hoiem, 2017) | ✓ | ✗ | 34.6 | 69.6 | 99.3 | 88.7 | 61.1 | 72.5 | 32.5 | 88.1 | 65.6 | 90.9 | 87.9 | 71.9 |
| iCaRL (Rebuffi et al., 2017) | ✓ | ✗ | 46.0 | 81.5 | 91.3 | 82.8 | 66.5 | 72.2 | 16.3 | 91.6 | 68.1 | 83.2 | 87.8 | 71.6 |
| LwF-VR (Ding et al., 2022) | ✓ | ✗ | 27.4 | 61.2 | 99.4 | 86.3 | 60.6 | 70.7 | 23.4 | 88.0 | 61.3 | 84.3 | **88.1** | 68.2 |
| WiSE-FT (Wortsman et al., 2022) | ✓ | ✗ | 35.6 | 76.9 | 99.5 | 89.1 | 62.1 | 71.8 | 27.8 | 90.8 | 67.0 | 85.6 | 87.6 | 72.2 |
| ZSCL (Zheng et al., 2023) | ✓ | ✗ | 78.2 | 91.1 | 97.6 | 92.5 | 87.4 | 78.2 | 45.0 | 92.3 | 72.7 | 96.2 | 86.3 | 83.4 |
| O-LoRA (Wang et al., 2023a) | ✓ | ✓ | 70.3 | 89.8 | 97.8 | 92.9 | 73.8 | 79.8 | 44.4 | 95.3 | 66.3 | 91.5 | 85.9 | 80.7 |
| InfLoRA (Liang & Li, 2024) | ✓ | ✓ | 82.4 | 92.0 | 99.3 | 93.9 | 85.4 | 81.2 | 46.1 | 96.5 | 70.0 | 97.6 | 87.2 | 84.7 |
| SD-LoRA (Wu et al., 2025b) | ✓ | ✓ | 72.3 | 89.7 | 97.3 | 92.4 | 76.1 | 78.9 | 45.3 | 95.2 | 61.6 | 96.9 | 86.1 | 81.1 |
| KeepLoRA | ✓ | ✓ | **83.7** | 92.2 | 99.5 | 94.4 | 90.7 | 81.3 | 49.0 | 96.9 | 72.3 | 98.0 | 87.3 | **85.9** |
| L2P (Wang et al., 2022c) | ✗ | ✓ | 80.1 | 89.1 | 99.1 | 93.8 | 96.2 | 76.5 | 40.1 | 86.9 | 73.5 | 86.3 | 84.2 | 82.3 |
| DualPrompt (Wang et al., 2022b) | ✗ | ✓ | 78.6 | 89.3 | 99.2 | 94.1 | 96.5 | 76.8 | 39.8 | 89.0 | 71.6 | 90.7 | 84.9 | 82.8 |
| S-Prompts (Wang et al., 2022a) | ✗ | ✓ | 79.2 | 89.1 | 99.1 | 94.3 | 95.8 | 76.3 | 39.9 | 95.5 | 70.1 | 97.6 | 84.4 | 83.8 |
| DIKI (Tang et al., 2024) | ✗ | ✓ | 81.9 | 89.2 | 99.4 | 94.3 | 96.8 | 76.7 | 46.3 | 95.9 | 74.8 | 98.3 | 86.6 | 85.5 |
| MoE-Adapters (Yu et al., 2024) | ✗ | ✗ | 84.1 | 88.5 | 94.0 | 91.8 | 94.1 | 77.8 | 50.4 | 93.3 | 77.1 | 87.7 | 86.6 | 84.1 |
| IAP (Fu et al., 2025) | ✗ | ✓ | 82.5 | 88.6 | 99.4 | 94.9 | 97.7 | 76.9 | 46.1 | 96.1 | 74.7 | 98.0 | 86.6 | 85.9 |
| KeepLoRA+ | ✗ | ✓ | **87.4** | 92.5 | 99.3 | 95.0 | 96.0 | 83.2 | 56.9 | 97.5 | 76.9 | 98.0 | 88.0 | 88.2 |

## C.3 PER-TRAINING-STEP RESULTS

We present the detailed per-training-step accuracies through all training steps in Tab. 7, 8, 9, 10, 11 and 12. These results demonstrate strong performance in terms of both learning plasticity and stability.

Table 7: Accuracy of KeepLoRA on the MTIL benchmark with order-I. Each row represents the performance on every dataset of the model trained after the corresponding task. Transfer , Average , and Last metrics are shown.

| | Aircraft | Caltech101 | CIFAR100 | DTD | EuroSAT | Flowers | Food | MNIST | OxfordPet | Cars | Sun397 | |
|---|---|---|---|---|---|---|---|---|---|---|---|---|
| Transfer | | 84.6 | 68.7 | 45.9 | 54.3 | 70.1 | 87.7 | 64.8 | 90.3 | 59.5 | 64.1 | 69.0 |
| Aircraft | 59.0 | 84.6 | 68.4 | 45.4 | 52.2 | 71.9 | 89.0 | 63.8 | 91.1 | 60.6 | 63.6 | |
| Caltech101 | 58.1 | 97.0 | 69.1 | 45.4 | 50.8 | 71.1 | 88.7 | 61.8 | 91.1 | 60.1 | 64.8 | |
| CIFAR100 | 56.0 | 96.8 | 87.6 | 46.8 | 56.3 | 68.9 | 87.3 | 66.3 | 90.1 | 59.6 | 64.7 | |
| DTD | 55.9 | 96.7 | 87.5 | 75.0 | 57.9 | 69.6 | 87.1 | 64.7 | 90.3 | 59.5 | 64.6 | |
| EuroSAT | 55.7 | 96.7 | 87.0 | 74.8 | 98.4 | 69.3 | 87.0 | 65.2 | 90.2 | 59.1 | 64.6 | |
| Flowers | 55.6 | 97.0 | 86.9 | 74.4 | 98.4 | 93.3 | 86.9 | 65.0 | 90.3 | 59.4 | 64.3 | |
| Food | 54.7 | 96.8 | 86.2 | 72.6 | 98.3 | 92.2 | 91.8 | 66.7 | 89.8 | 59.0 | 63.8 | |
| MNIST | 54.3 | 96.7 | 85.8 | 72.4 | 98.1 | 91.8 | 91.8 | 99.5 | 89.7 | 59.3 | 63.8 | |
| OxfordPet | 54.6 | 96.7 | 85.7 | 72.0 | 98.2 | 91.8 | 91.8 | 99.5 | 94.7 | 59.2 | 63.8 | |
| Cars | 54.2 | 96.7 | 85.7 | 71.9 | 98.1 | 91.5 | 91.7 | 99.5 | 94.4 | 84.3 | 63.7 | |
| SUN397 | 53.2 | 96.8 | 85.7 | 71.4 | 98.1 | 90.8 | 91.4 | 99.6 | 94.5 | 83.1 | 82.0 | 86.1 |
| Average | 55.6 | 95.7 | 83.2 | 65.6 | 82.2 | 82.0 | 89.5 | 77.4 | 91.5 | 63.9 | 65.8 | 77.5 |

Table 8: Accuracy of KeepLoRA on the MTIL benchmark with order-II. Each row represents the performance on every dataset of the model trained after the corresponding task. Transfer , Average , and Last metrics are shown.

| | Cars | Food | MNIST | OxfordPet | Flowers | Sun397 | Aircraft | Caltech101 | DTD | EuroSAT | CIFAR100 | |
|---|---|---|---|---|---|---|---|---|---|---|---|---|
| Transfer | | 88.7 | 57.7 | 91.2 | 72.1 | 65.8 | 23.4 | 88.8 | 45.4 | 48.5 | 68.2 | 65.0 |
| Cars | 86.2 | 88.7 | 57.1 | 91.3 | 71.7 | 65.5 | 23.5 | 87.4 | 46.6 | 50.7 | 69.5 | |
| Food | 85.9 | 92.9 | 58.3 | 91.1 | 72.3 | 66.0 | 23.9 | 88.3 | 45.3 | 49.8 | 70.5 | |
| MNIST | 85.8 | 92.8 | 99.6 | 91.2 | 71.9 | 66.2 | 23.0 | 88.6 | 46.4 | 50.4 | 68.1 | |
| OxfordPet | 85.7 | 92.8 | 99.6 | 94.8 | 72.4 | 65.9 | 23.0 | 89.3 | 46.0 | 48.2 | 67.8 | |
| Flowers | 85.6 | 92.8 | 99.6 | 94.8 | 92.4 | 65.7 | 23.0 | 89.3 | 46.2 | 46.9 | 67.5 | |
| Sun397 | 85.2 | 92.7 | 99.6 | 94.6 | 92.2 | 82.7 | 24.0 | 89.6 | 44.2 | 47.0 | 68.0 | |
| Aircraft | 84.8 | 92.7 | 99.6 | 94.6 | 92.1 | 82.7 | 51.6 | 89.3 | 44.2 | 46.3 | 68.0 | |
| Caltech101 | 84.8 | 92.7 | 99.6 | 94.6 | 92.2 | 82.6 | 51.6 | 97.1 | 44.5 | 48.7 | 68.3 | |
| DTD | 84.8 | 92.6 | 99.6 | 94.8 | 92.2 | 82.6 | 51.3 | 96.9 | 74.5 | 48.2 | 68.2 | |
| EuroSAT | 84.6 | 92.7 | 99.6 | 94.6 | 92.1 | 82.2 | 51.1 | 97.0 | 74.5 | 98.6 | 66.6 | |
| CIFAR100 | 83.7 | 92.3 | 99.5 | 94.4 | 90.8 | 81.3 | 49.0 | 96.9 | 72.3 | 98.0 | 87.3 | 85.9 |
| Average | 85.2 | 92.3 | 92.0 | 93.7 | 84.8 | 74.8 | 35.9 | 91.8 | 53.1 | 57.5 | 70.0 | 75.6 |

## USE OF LARGE LANGUAGE MODELS

We use the large language model to polish text and check grammar. All outputs were reviewed by the authors, who take full responsibility for the final content.

Table 9: Accuracy of KeepLoRA+ on the MTIL benchmark with order-I. Each row represents the performance on every dataset of the model trained after the corresponding task. Transfer, Average, and Last metrics are shown.

| | Aircraft | Caltech101 | CIFAR100 | DTD | EuroSAT | Flowers | Food | MNIST | OxfordPet | Cars | Sun397 | |
|---|---|---|---|---|---|---|---|---|---|---|---|---|
| Transfer | | 85.9 | 69.9 | 44.6 | 53.7 | 70.9 | 88.9 | 65.4 | 90.8 | 63.0 | 66.1 | 69.9 |
| Aircraft | 59.2 | 85.9 | 69.6 | 44.4 | 54.3 | 72.4 | 89.5 | 62.7 | 91.2 | 63.8 | 64.5 | |
| Caltech101 | 59.2 | 97.5 | 70.2 | 44.3 | 53.3 | 71.4 | 89.5 | 62.7 | 91.4 | 63.8 | 65.1 | |
| CIFAR100 | 59.0 | 97.8 | 88.2 | 45.1 | 52.7 | 70.3 | 88.7 | 67.8 | 90.8 | 63.3 | 66.3 | |
| DTD | 59.0 | 97.8 | 88.0 | 76.4 | 54.4 | 70.5 | 88.6 | 66.6 | 90.7 | 63.3 | 66.3 | |
| EuroSAT | 58.6 | 97.6 | 87.8 | 76.2 | 98.5 | 70.2 | 88.5 | 66.0 | 91.0 | 63.1 | 66.5 | |
| Flowers | 58.8 | 97.6 | 87.9 | 76.1 | 98.5 | 95.8 | 88.4 | 66.4 | 90.8 | 63.1 | 66.4 | |
| Food | 58.4 | 97.6 | 87.6 | 76.6 | 98.4 | 95.8 | 92.9 | 65.4 | 90.4 | 62.5 | 66.5 | |
| MNIST | 57.8 | 97.6 | 87.3 | 76.8 | 98.4 | 95.9 | 92.9 | 99.5 | 90.2 | 62.2 | 66.5 | |
| OxfordPet | 57.8 | 97.6 | 87.2 | 76.5 | 98.4 | 95.8 | 92.9 | 99.5 | 94.8 | 62.2 | 66.4 | |
| Cars | 57.7 | 97.5 | 87.3 | 76.7 | 98.4 | 95.6 | 92.9 | 99.5 | 94.8 | 87.7 | 66.3 | |
| SUN397 | 57.3 | 97.6 | 87.2 | 76.5 | 98.4 | 95.7 | 92.6 | 99.5 | 94.7 | 87.2 | 83.2 | 88.2 |
| Average | 58.4 | 96.5 | 84.4 | 67.8 | 82.1 | 84.5 | 90.7 | 77.8 | 91.9 | 67.5 | 67.6 | 79.0 |

Table 10: Accuracy of KeepLoRA+ on the MTIL benchmark with order-II. Each row represents the performance on every dataset of the model trained after the corresponding task. Transfer, Average, and Last metrics are shown.

| | Cars | Food | MNIST | OxfordPet | Flowers | Sun397 | Aircraft | Caltech101 | DTD | EuroSAT | CIFAR100 | |
|---|---|---|---|---|---|---|---|---|---|---|---|---|
| Transfer | | 89.1 | 58.1 | 90.7 | 72.4 | 65.4 | 24.0 | 88.9 | 44.0 | 52.7 | 70.2 | 65.5 |
| Cars | 88.4 | 89.1 | 58.9 | 91.6 | 72.3 | 65.3 | 24.2 | 88.0 | 44.8 | 53.9 | 70.1 | |
| Food | 88.3 | 92.8 | 57.2 | 90.1 | 72.4 | 65.4 | 24.2 | 88.5 | 44.5 | 52.5 | 70.8 | |
| MNIST | 88.1 | 92.8 | 99.4 | 90.4 | 72.1 | 65.5 | 24.1 | 88.5 | 44.4 | 52.9 | 70.1 | |
| OxfordPet | 88.3 | 92.7 | 99.4 | 95.2 | 72.8 | 65.3 | 24.1 | 88.7 | 44.2 | 52.2 | 70.2 | |
| Flowers | 88.3 | 92.8 | 99.5 | 95.2 | 96.1 | 65.3 | 23.9 | 88.7 | 44.4 | 51.7 | 69.9 | |
| Sun397 | 87.9 | 92.7 | 99.4 | 95.0 | 96.1 | 83.5 | 23.6 | 89.9 | 43.4 | 52.8 | 70.2 | |
| Aircraft | 87.9 | 92.7 | 99.4 | 95.0 | 96.1 | 83.5 | 57.8 | 90.1 | 43.1 | 52.8 | 70.2 | |
| Caltech101 | 87.7 | 92.7 | 99.4 | 95.0 | 96.0 | 83.5 | 57.5 | 97.4 | 43.0 | 52.6 | 70.4 | |
| DTD | 87.7 | 92.7 | 99.4 | 95.1 | 96.0 | 83.4 | 57.4 | 97.4 | 76.3 | 52.7 | 70.1 | |
| EuroSAT | 87.6 | 92.7 | 99.5 | 95.0 | 95.9 | 83.4 | 57.2 | 97.4 | 76.3 | 98.3 | 70.3 | |
| CIFAR100 | 87.4 | 92.5 | 99.3 | 95.0 | 96.0 | 83.3 | 56.9 | 97.5 | 76.9 | 98.0 | 88.0 | 88.2 |
| Average | 88.0 | 92.4 | 91.9 | 93.9 | 87.4 | 75.2 | 39.2 | 92.0 | 52.8 | 60.9 | 71.8 | 76.9 |

Table 11: Accuracy of LoRA-FT, O-LoRA, CL-MoE, SEFE, KeepLoRA on the MLLM-DCL benchmark. Each row represents the performance on every dataset of the model trained after the corresponding task. Transfer , Average , and Last metrics are shown.

(a) LoRA-FT

|  | Sensing | Medical | Driving | Science | Finance | Metric |
|---|---|---|---|---|---|---|
| Transfer |  | 28.1 | 17.4 | 34.0 | 50.2 | 32.4 |
| Sensing | 78.8 | 28.1 | 17.3 | 34.8 | 55.6 | |
| Medical | 75.5 | 58.4 | 17.5 | 32.7 | 54.8 | |
| Driving | 70.0 | 47.5 | 52.3 | 34.6 | 40.9 | |
| Science | 73.2 | 46.4 | 40.6 | 50.4 | 49.5 | |
| Finance | 69.3 | 44.3 | 29.1 | 41.4 | 88.4 | 54.5 |
| Average | 73.3 | 44.9 | 31.4 | 38.8 | 57.8 | 49.3 |

(b) O-LoRA

|  | Sensing | Medical | Driving | Science | Finance | Metric |
|---|---|---|---|---|---|---|
| Transfer |  | 28.4 | 18.4 | 33.7 | 52.5 | 33.3 |
| Sensing | 79.4 | 28.4 | 17.6 | 34.9 | 56.1 | |
| Medical | 74.3 | 58.5 | 19.2 | 33.2 | 56.0 | |
| Driving | 74.7 | 48.3 | 52.6 | 33.1 | 45.2 | |
| Science | 74.6 | 46.5 | 42.2 | 50.1 | 52.8 | |
| Finance | 72.3 | 46.9 | 31.6 | 41.5 | 88.1 | 56.1 |
| Average | 75.0 | 45.7 | 32.6 | 38.5 | 59.6 | 50.3 |

(c) CL-MoE

|  | Sensing | Medical | Driving | Science | Finance | Metric |
|---|---|---|---|---|---|---|
| Transfer |  | 28.3 | 19.4 | 34.1 | 48.6 | 32.6 |
| Sensing | 79.4 | 28.3 | 18.7 | 35.2 | 56.4 | |
| Medical | 74.8 | 60.7 | 20.1 | 32.4 | 54.9 | |
| Driving | 74.0 | 44.3 | 52.1 | 34.7 | 39.6 | |
| Science | 71.0 | 47.4 | 40.0 | 50.7 | 43.3 | |
| Finance | 71.8 | 47.4 | 29.5 | 41.5 | 89.2 | 55.9 |
| Average | 74.2 | 45.6 | 32.1 | 38.9 | 56.7 | 49.5 |

(d) SEFE

|  | Sensing | Medical | Driving | Science | Finance | Metric |
|---|---|---|---|---|---|---|
| Transfer |  | 28.1 | 19.6 | 33.9 | 52.4 | 33.5 |
| Sensing | 78.8 | 28.1 | 18.6 | 35.1 | 56.2 | |
| Medical | 77.1 | 59.5 | 20.7 | 33.0 | 55.7 | |
| Driving | 77.8 | 51.6 | 52.5 | 33.5 | 47.4 | |
| Science | 77.9 | 48.4 | 44.7 | 50.4 | 50.1 | |
| Finance | 77.1 | 50.9 | 40.3 | 43.0 | 88.4 | 59.9 |
| Average | 77.7 | 47.7 | 35.4 | 39.0 | 59.6 | 51.9 |

(e) KeepLoRA

|  | Sensing | Medical | Driving | Science | Finance | Metric |
|---|---|---|---|---|---|---|
| Transfer |  | 28.5 | 16.6 | 34.1 | 55.6 | 33.7 |
| Sensing | 80.0 | 28.5 | 17.0 | 35.1 | 55.1 | |
| Medical | 79.9 | 58.6 | 16.3 | 33.7 | 55.6 | |
| Driving | 79.8 | 57.7 | 53.1 | 33.7 | 54.6 | |
| Science | 79.2 | 54.9 | 51.1 | 51.6 | 57.2 | |
| Finance | 78.8 | 54.3 | 50.2 | 49.5 | 89.3 | 64.4 |
| Average | 79.6 | 50.8 | 37.5 | 40.7 | 62.4 | 54.2 |

Table 12: Accuracy of LoRA-FT, O-LoRA, CL-MoE, SEFE, KeepLoRA on the UCIT benchmark. Each row represents the performance on every dataset of the model trained after the corresponding task. Transfer , Average , and Last metrics are shown.

(a) LoRA-FT

| | ImgNet-R | ArxivQA | VizWiz | IconQA | CLEVR | Flickr30k | |
|---|---|---|---|---|---|---|---|
| Transfer | | 52.6 | 18.3 | 6.0 | 17.0 | 40.3 | 26.8 |
| ImgNet-R | 91.7 | 52.6 | 23.5 | 11.8 | 17.2 | 36.5 | |
| ArxivQA | 90.5 | 92.1 | 13.1 | 2.1 | 14.2 | 21.5 | |
| VizWiz | 73.6 | 90.7 | 61.0 | 4.2 | 19.0 | 49.7 | |
| IconQA | 72.7 | 77.1 | 53.7 | 79.7 | 17.4 | 47.8 | |
| CLEVR | 68.8 | 77.4 | 52.3 | 67.8 | 77.9 | 46.1 | |
| Flickr30k | 58.6 | 76.7 | 45.7 | 67.4 | 61.6 | 58.0 | 61.4 |
| Average | 76.0 | 77.8 | 41.6 | 38.8 | 34.6 | 43.3 | 52.0 |

(b) O-LoRA

| | ImgNet-R | ArxivQA | VizWiz | IconQA | CLEVR | Flickr30k | |
|---|---|---|---|---|---|---|---|
| Transfer | | 52.9 | 19.6 | 4.4 | 16.9 | 41.0 | 27.0 |
| ImgNet-R | 91.5 | 52.9 | 24.7 | 13.3 | 17.3 | 36.5 | |
| ArxivQA | 89.7 | 94.2 | 14.5 | 0.0 | 12.9 | 25.0 | |
| VizWiz | 80.9 | 91.7 | 59.8 | 0.0 | 19.6 | 49.0 | |
| IconQA | 80.2 | 80.3 | 54.5 | 75.9 | 17.6 | 48.6 | |
| CLEVR | 78.1 | 80.4 | 51.6 | 63.2 | 72.4 | 46.0 | |
| Flickr30k | 74.2 | 80.9 | 45.3 | 62.9 | 63.8 | 57.2 | 64.1 |
| Average | 82.4 | 80.1 | 41.7 | 35.9 | 33.9 | 43.7 | 53.0 |

(c) CL-MoE

| | ImgNet-R | ArxivQA | VizWiz | IconQA | CLEVR | Flickr30k | |
|---|---|---|---|---|---|---|---|
| Transfer | | 52.0 | 19.3 | 7.4 | 17.8 | 41.3 | 27.6 |
| ImgNet-R | 91.2 | 52.0 | 23.9 | 5.2 | 15.6 | 36.9 | |
| ArxivQA | 89.2 | 92.5 | 14.8 | 10.0 | 15.7 | 26.2 | |
| VizWiz | 77.2 | 90.7 | 60.4 | 6.9 | 20.6 | 49.5 | |
| IconQA | 79.5 | 76.2 | 51.0 | 54.7 | 19.4 | 47.9 | |
| CLEVR | 76.7 | 75.4 | 48.1 | 52.6 | 73.0 | 45.9 | |
| Flickr30k | 61.2 | 75.8 | 44.4 | 52.6 | 54.4 | 57.3 | 58.6 |
| Average | 80.2 | 77.1 | 40.4 | 30.3 | 33.1 | 44.0 | 50.9 |

(d) SEFE

| | ImgNet-R | ArxivQA | VizWiz | IconQA | CLEVR | Flickr30k | |
|---|---|---|---|---|---|---|---|
| Transfer | | 53.3 | 18.7 | 7.5 | 17.0 | 40.9 | 27.5 |
| ImgNet-R | 91.6 | 53.3 | 23.7 | 12.1 | 16.9 | 36.4 | |
| ArxivQA | 90.4 | 92.8 | 13.7 | 5.0 | 16.4 | 21.1 | |
| VizWiz | 83.6 | 89.3 | 61.4 | 5.3 | 18.6 | 49.8 | |
| IconQA | 84.3 | 78.1 | 57.4 | 79.6 | 16.2 | 50.6 | |
| CLEVR | 82.8 | 78.6 | 54.2 | 70.6 | 75.0 | 46.5 | |
| Flickr30k | 80.2 | 79.1 | 47.1 | 69.4 | 65.7 | 57.3 | 66.5 |
| Average | 85.5 | 78.6 | 42.9 | 40.3 | 34.8 | 43.6 | 54.3 |

(e) KeepLoRA

| | ImgNet-R | ArxivQA | VizWiz | IconQA | CLEVR | Flickr30k | |
|---|---|---|---|---|---|---|---|
| Transfer | | 52.8 | 20.4 | 9.2 | 18.1 | 41.5 | 28.4 |
| ImgNet-R | 91.5 | 52.8 | 25.6 | 13.4 | 17.1 | 36.7 | |
| ArxivQA | 90.4 | 94.5 | 15.2 | 4.0 | 17.2 | 21.5 | |
| VizWiz | 85.5 | 92.4 | 61.5 | 10.1 | 21.0 | 50.6 | |
| IconQA | 85.1 | 86.0 | 55.7 | 76.9 | 17.1 | 50.9 | |
| CLEVR | 84.1 | 89.3 | 51.5 | 68.3 | 72.6 | 47.8 | |
| Flickr30k | 82.4 | 86.7 | 46.6 | 67.8 | 66.4 | 57.2 | 67.8 |
| Average | 86.5 | 83.6 | 42.7 | 40.1 | 35.2 | 44.1 | 55.4 |

