# OpenReview forum: "KeepLoRA: Continual Learning with Residual Gradient Adaptation"
_ICLR.cc/2026/Conference — ICLR 2026 Poster_

### Official Review · Reviewer_gbxX · 2025-10-26

**Soundness:** 4
**Presentation:** 4
**Contribution:** 3
**Rating:** 6
**Confidence:** 2

**Summary:**

This paper focuses on balancing three core objectives in continual learning (CL) for pre-trained vision-language models (VLMs): preserving pre-trained general knowledge, retaining sequential task knowledge, and maintaining new task plasticity. Via SVD analysis, it finds general knowledge resides in the parameter principal subspace, while task-specific knowledge is in the residual subspace. Based on this, KeepLoRA restricts LoRA updates to the residual subspace—by projecting new task gradients onto the subspace orthogonal to the pre-trained principal subspace and previous task dominant directions—and initializes LoRA via purified gradients. Theoretical proofs and MTIL benchmark experiments (11 image classification tasks) confirm its state-of-the-art performance on Transfer, Average, and Last metrics.

**Strengths:**

- Insightful subspace analysis: The separation of general/task-specific knowledge across principal/residual subspaces provides a clear, interpretable basis for CL design.

- Theoretical rigor: Propositions 3.1 and 3.2 mathematically validate that KeepLoRA’s updates ensure stability and align with optimal plasticity.

- Practicality: Parameter-efficient (only updates LoRA’s B matrix, no inference overhead), data-independent (no replay/reference data), and robust across task orders.

- Comprehensive evaluation: Uses well-defined metrics (Transfer/Average/Last) and ablation studies to verify component contributions.

**Weaknesses:**

1. KeepLoRA’s design assumes that "residual subspace updates for new tasks do not interfere with old tasks," but it does not analyze how task similarity affects this assumption: For highly similar tasks (e.g., two fine-grained classification tasks like StanfordCars and Aircraft), their residual subspaces may overlap, potentially leading to interference.

2. Scalability gaps: Unaddressed feasibility of SVD and subspace scaling for large VLMs (e.g., CLIP-L) with massive parameters.

**Questions:**

1. Some typographical errors. For example, "KeppLoRA" in table 2 and row 816.

---

> ### Author Response · Authors · 2025-11-17
>
> We appreciate very much your constructive comments on our paper. We address your concerns one by one in the followings.
>
> > **W1**: KeepLoRA’s design assumes that "residual subspace updates for new tasks do not interfere with old tasks," but it does not analyze how task similarity affects this assumption: For highly similar tasks (e.g., two fine-grained classification tasks like StanfordCars and Aircraft), their residual subspaces may overlap, potentially leading to interference.
>
> Our design ensures that general knowledge is encoded in the principal subspace and remains frozen. Each task aims to learn its distinct parts in the residual subspace. Even for highly similar tasks, the parameter redundancy in deep neural networks allows us to allocate completely orthogonal spaces for their respective updates within the residual subspace. The orthogonality guarantees that tasks do not interfere with each other, regardless of their similarity. Empirically, Tables 7 and 8 demonstrate KeepLoRA's performance on the MTIL benchmark with two different task orders, showing no catastrophic forgetting for similar tasks such as StanfordCars and Aircraft. Furthermore, Section 4.2 Analysis of Plasticity and Fig. 2 illustrate that KeepLoRA effectively learns new tasks within their respective orthogonal residual subspaces.
>
> > **W2**: Scalability gaps: Unaddressed feasibility of SVD and subspace scaling for large VLMs (e.g., CLIP-L) with massive parameters.
>
> We acknowledge the importance of demonstrating KeepLoRA's scalability on models with larger parameter counts. While CLIP ViT-B/16 already achieves high accuracy, over 90% on 6 of the 11 MTIL image classification datasets, proving its sufficiency for these tasks. We have further extended our evaluation to **vision question answering** tasks using **LLaVA-1.5-7b**. These new experiments, conducted on the MLLM-DCL and UCIT continual learning benchmarks, demonstrate KeepLoRA's effectiveness and scalability on models with substantially larger parameter numbers. **Please refer to our main Official Comment for these results, with further details available in the revised paper (Tables 3, 4 and Appendix B.1 and B.4).**
>
> > Q1: Some typographical errors. For example, "KeppLoRA" in table 2 and row 816.
>
> Thanks for pointing out these typographical errors. They have been corrected in the revised manuscript.

---

> > ### Comment · Reviewer_gbxX · 2025-11-26
> >
> > Thank you for your detailed reply, which resolved my main questions. I will maintain a positive rating.

---

> > > ### Author Response · Authors · 2025-11-26
> > >
> > > Thank you very much for your response! We truly appreciate your positive feedback. We remain available for further discussion if you have any additional questions.

---

### Official Review · Reviewer_1c8k · 2025-10-29

**Soundness:** 3
**Presentation:** 3
**Contribution:** 3
**Rating:** 4
**Confidence:** 5

**Summary:**

This paper introduces KeepLoRA, a novel method for continual learning (CL) in pre-trained vision-language models designed to balance plasticity, backward stability, and forward stability. The authors of this study found a key insight by looking at how a model stores information, discovering that general, pre-trained knowledge is stored in the principal subspace, while specific skills are stored in the residual subspace. Motivated by this, KeepLoRA learns new tasks by restricting its Low-Rank Adaptation (LoRA) parameter updates to this residual subspace, where it essentially keeps the parameters from the principal subspace and everything it learned from past tasks, ensuring the new information is written only into the residual subspace. This approach prevents interference with existing knowledge, and the paper's theoretical and empirical analyses confirm that KeepLoRA effectively balances the three CL objectives to achieve state-of-the-art performance.

**Strengths:**

1. State of the Art performance: The paper demonstrates that KeepLoRA and its variant, KeepLoRA+, achieve state-of-the-art results on the MTIL benchmark, outperforming previous methods on all key metrics

2. Strong Empirical Analysis: The paper's core hypothesis is based on a clear, intuitive analysis of the model's parameter space. The finding that general knowledge resides in the principal subspace while task-specific knowledge is in the residual subspace provides a solid foundation for the method.

3.Theoretically Grounded: The authors provide a strong theoretical justification for their approach. They prove that their method of initializing and freezing the LoRA matrix At is an optimal solution (Proposition 3.2) to the problem of maximizing adaptation to the new task while remaining perfectly orthogonal to all previously learned knowledge.

**Weaknesses:**

1. Limited Evaluation on Language: The paper focuses on Vision-Language Models (VLMs), but the evaluation is performed on a benchmark of 11 image classification datasets. The "language" aspect is only used for zero-shot classification via class names. The method's effectiveness on more complex, language-heavy VLM tasks is unproven.

2. Your method, by design, projects new task gradients to be orthogonal to the principal subspace and all previous task directions to ensure stability. Does this strict orthogonality constraint inadvertently prevent positive forward transfer, where the model should be re-using and building upon those similar, previously-learned features, rather than being forced to find a completely new, orthogonal direction to learn?

**Questions:**

see weaknesses

---

> ### Author Response · Authors · 2025-11-17
>
> We appreciate very much your constructive comments on our paper. We address your concerns one by one in the followings.
>
> > **W1: Limited Evaluation on Language**: The paper focuses on Vision-Language Models (VLMs), but the evaluation is performed on a benchmark of 11 image classification datasets. The "language" aspect is only used for zero-shot classification via class names. The method's effectiveness on more complex, language-heavy VLM tasks is unproven.
>
> We acknowledge the limitation in our initial submission. We have conducted experiments on more complex, language-heavy VLM tasks using **LLaVA-1.5-7b** on two recently proposed **vision question answering** continual learning benchmarks **MLLM-DCL** and **UCIT**. **Please refer to our main Official Comment for these results, with further details available in the revised paper (Tables 3, 4 and Appendix B.1 and B.4).**
>
> > **W2**: Your method, by design, projects new task gradients to be orthogonal to the principal subspace and all previous task directions to ensure stability. Does this strict orthogonality constraint inadvertently prevent positive forward transfer, where the model should be re-using and building upon those similar, previously-learned features, rather than being forced to find a completely new, orthogonal direction to learn?
>
> Thank you for this insightful question. The strict orthogonality constraint is indeed deliberate, which guarantees minimal interference with the principal subspace of pre-trained knowledge (forward stability) and the dominant directions of previous tasks (backward stability). Without a hard constraint, even minor drift along protected directions causes cumulative forgetting, as seen in prior LoRA-based methods.
>
> **Does this strict orthogonality constraint enable the model to learn new tasks effectively?**
>
> **Yes.** Our experiments in Section 4.2 Analysis of Plasticity and Fig. 2 demonstrate that KeepLoRA learns new tasks effectively even when updates are confined to the residual subspace. It is because general features essential for new tasks are already robustly encoded within the protected principal subspace, allowing new tasks to freely reuse them without needing to update those directions. Leveraging the inherent parameter redundancy in large models, we can still find lottery ticket networks [1] within the residual subspace for further performance enhancement.
>
> **Does this strict orthogonality constraint improve performance on previously learned or unseen tasks via positive transfer?**
>
> **Not directly, and not our primary design goal.** As elaborated in our motivation and paper, KeepLoRA is specifically designed to maintain stability and plasticity rather than to enhance positive forward transfer in an undirected manner. Achieving such an improvement would be complex: it would require not only updating specific parameters but doing so in a direction that simultaneously boosts transfer performance without compromising current task performance. This entails precisely identifying which knowledge from the current task is truly beneficial for broader transfer, a challenging problem potentially related to model interpretability or knowledge distillation. We consider this a truly valuable and promising direction for future work!
>
> [1] Frankle J, Carbin M. The Lottery Ticket Hypothesis: Finding Sparse, Trainable Neural Networks[C]//International Conference on Learning Representations. 2019.

---

> ### Author Response · Authors · 2025-11-27
>
> Dear Reviewer 1c8k,
>
> Thank you very much for your valuable comments during the review process. We have carefully answered your concerns by providing additional explanations and supporting experimental results in our response.
> We sincerely hope that our reply adequately resolves your questions. If you have further inquiries or suggestions, we would be more than happy to continue the discussion and provide any additional information needed.
>
> Thank you again for your time and effort in reviewing our work.
>
> Best regards,
>
> The Authors

---

> > ### Comment · Reviewer_1c8k · 2025-11-27
> >
> > The reviewer appreciates the author's feedback and adjusts the score accordingly.
> >
> > However, I still want to discuss with the authors how this strict orthogonality constraint can improve performance on previously learned or unseen tasks via positive transfer. I think this will be very interesting.

---

> > > ### Author Response · Authors · 2025-11-28
> > >
> > > We sincerely thank the reviewer for raising the score and for your continued engagement with our work.
> > >
> > > Regarding your query, **this strict orthogonality constraint is not primarily designed to directly improve performance on previously learned or unseen tasks via positive transfer.** Instead, its core objective is to preserve both the knowledge acquired during continual learning and the original pre-trained knowledge by enforcing orthogonality to the principal subspace.
> > >
> > > This design yields significant improvements over various baselines on unseen tasks. As shown in Tab. 2 on the MTIL benchmark, the zero-shot performance on unseen tasks (Transfer metric) represents a **1.6% improvement** compared to the current state-of-the-art method (under the Arch. Kept and w/o Extra Data setting) while exhibits a negligible degradation of only **0.41%** of vanilla CLIP.
> > >
> > > Actually, we indeed observed that for certain tasks, the transfer capability improved compared to the zero-shot baseline. Specifically, **MNIST showed an average improvement of 5.4%, and DTD improved by 1.3%.** We attribute it to the fact that strictly orthogonalizing against the principal weight subspace ensures that the model's general knowledge remains almost intact. By updating only within the residual subspace, the model acquires task-specific skills without corrupting its general foundation. **The preservation makes it possible to achieve transfer gains on specific datasets**, as observed in our results. A similar effect applies to the performance improvement on previously learned tasks.
> > >
> > > In summary, KeepLoRA is founded on the observation that the principal subspace (spanned by components with large singular values) predominantly encodes general knowledge, while the residual subspace (spanned by small singular values) encodes domain-specific knowledge. We utilize strict orthogonality to prevent forgetting. The instances of positive transfer we observed suggest that minimizing interference via KeepLoRA can indeed facilitate improvements in certain general or specific domains.
> > >
> > > Achieving stable positive transfer across all datasets is an desirable direction for future work, but we consider this must be based on two prerequisites:
> > >
> > > 1. The ability to correctly identify which samples or gradient information are beneficial for positive transfer.
> > > 2. Determining exactly which parts of the parameters to modify and how to modify to enhance capability without causing forgetting.
> > >
> > > Thank you again for your insightful question regarding positive transfer. We believe it is a direction worth further exploration based on the KeepLoRA.

---

### Official Review · Reviewer_FhjQ · 2025-10-30

**Soundness:** 3
**Presentation:** 3
**Contribution:** 3
**Rating:** 6
**Confidence:** 4

**Summary:**

This paper analyses the parameter space using SVD and finds that the principal subspace encodes general knowledge and the residual subspace encodes domain-specific knowledge. The paper proposes KeepLoRa which projects the gradients onto a subspace
orthogonal to both the principal subspace of pre-trained model and the dominant directions of previous task features. Theoretical analysis and empirical results on MTIL are provided.

**Strengths:**

1. KeepLoRa works with pre-trained models addressing three competing objectives: maintaining the ability to learn new knowledge (plasticity), preventing the forgetting of previously learned tasks (backward stability), and preserving the general pre-trained knowledge that guarantees general transferability (forward stability).
2. KeepLora can be implemented in a relatively straightforward manner.
3. Sensible theoretical analysis and strong empirical performance.

**Weaknesses:**

1. KeepLora+ outperforms KeepLora but the first mention of KeepLora+ is in Table 2 and $4.1.
2. KeepLora requires storing dominant singular vectors from tasks M but this is not listed in Table 2 nor analysed elsewhere.
3. KeepLora introduces epsilon_w, epsilon_f, r, alpha hyper parameters but only the sensitivity of epsilon_w(vision) and epsilon_w(text) is presented.

**Questions:**

1. What is the computational and memory overhead KeepLoRa and keepLoRa+?
2. What is the sensitivity of all hyper parameters?

---

> ### Author Response · Authors · 2025-11-18
> **Part I**
>
> We appreciate very much your constructive comments on our paper. We address your concerns one by one in the followings.
>
> > **W1**: KeepLora+ outperforms KeepLora but the first mention of KeepLora+ is in Table 2 and $4.1.
>
> Thank you for pointing out writing issue. We have revised the manuscript to address it. KeepLoRA+ is now explicitly introduced in the text of Section 4.1, with a reference to its implementation details in Appendix B.3. The core contribution of our paper is KeepLoRA, which is derived from our subspace analysis. KeepLoRA+ is presented as a variant for classification tasks to demonstrate that the foundational KeepLoRA method is simple, effective, and can be readily extended for further performance gains.
>
>
>
> > **W2**: KeepLora requires storing dominant singular vectors from tasks M but this is not listed in Table 2 nor analysed elsewhere.
>
> Thank you for this question. We believe there might be a small misunderstanding regarding the table reference. You may be referring to Algorithm 1, as Table 2 presents experimental results.
>
> While $M$ is not explicitly written as a variable in every line of Algorithm 1, its usage is implicitly included through the references to the equations. Specifically $M$ is used in Eq. 5 for the gradient projection during LoRA initialization referenced in Line 4 of the algorithm. And, the update mechanism for $M$ is detailed in Eq. 3 and 4 referenced in Line 8 of the algorithm. A detailed description and formal definition of $M$ are provided in the text accompanying these equations.
>
> To further elaborate on its importance, we provide a deeper analysis of how the design of $M$ mitigates catastrophic forgetting below:
>
> Let $X \in \mathbb{R}^{d \times n}$ denote the layer feature matrix of a task. We define $M \in \mathbb{R}^{d \times p}$ as the top-$p$ principal components of $X$, which capture at least an $\epsilon_f$ fraction of the total variance, satisfying$\|\|M M^{\top} X \|\|_{F}^{2} \ge  \epsilon_f \|\| X \|\|_F^{2}$.
>
> The feature matrix $X$ can then be decomposed as: $ X = M M^{\top} X + (I-MM^{\top})X$,  where $\|\|(I-MM^{\top})X\|\|_{F} ^{2} = \|\| X\|\|_F^{2} -   \|\|M M^{\top} X \|\|_F^{2} \le (1-  \epsilon_f) \|\|X\|\|_F^{2}$ .
>
> Consider an incremental parameter update $\Delta W \in \mathbb{R}^{d \times d}$ when learning a new task. If we enforce that $\Delta W$ is orthogonal to the principal subspace, $\Delta W M = 0$, then: $ \Delta WX =   \Delta W \left(M M^{\top} X + (I-MM^{\top})X\right) = \Delta W (I-MM^{\top})X$.
>
> Therefore, the interference induced on the features satisfies: $\|\|\Delta W X\|\|_F^{2} = \|\|\Delta W (I-MM^{\top})X\|\|_F^{2} \le (1 - \epsilon_f)\|\|\Delta W \|\|_F^{2} \|\| X\|\|_F^{2}$.
>
> Since the residual $(I - MM^{\top})X$ contains at most $(1 - \epsilon_f)$ fraction of the total variance of $X$, the update $\Delta W$ only acts on low-variance directions. Consequently, the induced change $\|\|\Delta W X\|\|_F^2$ remains small, ensuring that the learned task representations are minimally perturbed, thereby mitigating catastrophic forgetting.
>
> Empirically, we obtain $M$ by performing SVD on the feature matrix $X = U\Sigma V^{\top}$. The principal directions of $X$ correspond to the left singular vectors associated with the largest singular values, whose squared magnitudes represent the variance captured by each component. We select the smallest $p$ such that the top-$p$ singular values account for at least an $\epsilon_f$ fraction of the total variance, i.e., $\sum_{i=1}^{p}\sigma_i^2 \geq \epsilon_f \sum_{i}\sigma_i^2$, and form $M$ by taking the first $p$ columns of $U$. This ensures that $MM^{\top}X$ retains at least an $\epsilon_f$ proportion of the energy in $X$.

---

> > ### Author Response · Authors · 2025-11-18
> > **Part II**
> >
> > > **W3**: KeepLora introduces epsilon_w, epsilon_f, r, alpha hyper parameters but only the sensitivity of epsilon_w(vision) and epsilon_w(text) is presented.
> > >
> > > **Q2**: What is the sensitivity of all hyper parameters?
> >
> > Thank you for this suggestion. We have conducted additional experiments to analyze the sensitivity of $\epsilon_f$and $r$, supplementing the existing analysis of $\epsilon_w$ in Appendix C.2.
> >
> > Analysis of $\epsilon_f$:
> >
> > | $\epsilon_f$ | 0.9  | 0.95 | 0.99 | 0.995 |
> > | ------------ | ---- | ---- | ---- | ----- |
> > | Last         | 81.3 | 83.0 | 86.1 | 86.2  |
> >
> > The result shows that the performance improves as $\epsilon_f$ increases, stabilizing at high values $\geq$ 0.99, which indicates that preserving a high fraction of the feature energy is critical for maintaining the stability of previously learned tasks. Based on this, we set $\epsilon_f$=0.99 in our experiments as a robust choice.
> >
> > Analysis of LoRA rank $r$:
> >
> > | $r$      | 8    | 16   | 32   | 64   |
> > | -------- | ---- | ---- | ---- | ---- |
> > | Transfer | 67.8 | 69.0 | 69.4 | 69.8 |
> > | Average  | 76.6 | 77.5 | 77.9 | 78.0 |
> > | Last     | 84.9 | 86.1 | 86.6 | 86.7 |
> >
> > Performance steadily improves with a larger rank $r$, showing robustness across a range of values.
> >
> > For the scaling factor $\alpha$, we consistently set it to 1 across all experiments to maintain a small update magnitude. We set r=16 for KeepLoRA in our main comparisons to ensure a fair comparison with other LoRA-based methods under a similar parameter budget. We will include this full analysis in the revised appendix.
> >
> >
> >
> > > **Q1**: What is the computational and memory overhead KeepLoRa and keepLoRa+?
> >
> > We benchmarked the overhead of our methods against relevant baselines. The table below summarizes the trainable parameters, maximum GPU memory usage, and training time per batch with batch size=64. Time comparisons for MoE-Adapters, KeepLoRA, and KeepLoRA+ were conducted on a single NVIDIA 4090 GPU.
> >
> > | Method       | Train Params (M) | Max GPU Memory (GB) | Time (s/batch) |
> > | :----------- | :--------------- | :------------------ | :------------- |
> > | LWF          | 149.6            | 31.42               | --             |
> > | ZSCL         | 149.6            | 25.67               | --             |
> > | MoE-Adapters | 59.8             | 21.83               | 0.337          |
> > | KeepLoRA     | 0.98             | 19.79               | 0.222          |
> > | KeepLoRA+    | 1.04             | 19.89               | 0.224          |
> >
> > As shown, KeepLoRA is significantly more efficient than full fine-tuning (LWF, ZSCL) and architecture-extension (MoE-Adapters) methods in terms of trainable parameters, memory, and speed. Since KeepLoRA retains the standard LoRA structure for backpropagation, its training overhead is comparable to vanilla LoRA. The additional cost for KeepLoRA+ is minimal, as it only introduces a small prototype vector for each class.
> >
> >
> >
> > In addition to the points above, we would like to highlight that we have scaled KeepLoRA to the **LLaVA-1.5-7B** model for vision question answering tasks. **Please refer to our main Official Comment for these results, with further details available in the revised paper (Tables 3, 4 and Appendix B.1 and B.4).**

---

### Official Review · Reviewer_zp4o · 2025-10-31

**Soundness:** 3
**Presentation:** 3
**Contribution:** 3
**Rating:** 6
**Confidence:** 4

**Summary:**

The paper addresses continual learning (CL) in pre-trained vision-language models (VLMs) by balancing plasticity and stability. It posits that general knowledge resides in the principal subspace of model weights, while task-specific knowledge lies in the residual subspace. Building on this idea, KeepLoRA confines new task updates to the residual subspace by projecting gradients orthogonally to a unified principal subspace and initializing a frozen LoRA adapter. Supported by theoretical analysis, KeepLoRA achieves comparable performance on the MTIL benchmark.

**Strengths:**

1. Clear  idea of separating general (principal) and specific (residual) knowledge in the parameter space.


2. Provides strong theoretical justification connecting the method to optimal, constrained gradient descent.

3. Good practicality as it adds no inference overhead, unlike architecture-extension methods.

**Weaknesses:**

1. Unclear if the unified principal subspace, which accumulates past task directions, can scale to a large number of tasks without prohibitive cost.

2. The method requires expensive per-task full-gradient computation and a one-time full-model SVD, which are not fully benchmarked.

3. Relies on crucial hyperparameters (e.g., $\epsilon_w$, $\epsilon_f$) whose robustness and sensitivity are not deeply analyzed.

**Questions:**

Refer to Weaknesses for related questions.

---

> ### Author Response · Authors · 2025-11-18
> **Part I**
>
> We appreciate very much your constructive comments on our paper. We address your concerns one by one in the followings.
>
> > **W1**: Unclear if the unified principal subspace, which accumulates past task directions, can scale to a large number of tasks without prohibitive cost.
>
> Yes, our method can scale to a large number of tasks without prohibitive cost because the update process for the principal subspace of each task relies only on its previous state and the current task's dominant directions.
>
> As outlined in Algorithm 1, for each new task, we perform two main steps. Line 4 LoRA Initialization: We project the initial gradient to initialize LoRA adapters, as defined in Eq. 5 and 6. Line 8 Subspace Update: We extract dominant feature directions to update the principal subspace, following Eq. 3 and 4. The complexity of these steps depends on the feature dimension and the rank of the update, not on the accumulated number of past tasks. And, the total dimension of the unified principal subspace is upper-bounded by the feature dimension d_in, ensuring that the projection and SVD operations remain efficient and do not scale with an increasing number of tasks. We have clarify this in the revised version.
>
>
>
> > **W2**: The method requires expensive per-task full-gradient computation and a one-time full-model SVD, which are not fully benchmarked.
>
> Thank you for the important question. We would like to clarify the scope of our computations and provide a quantitative breakdown of the additional overhead below.
>
> The gradient computation and subsequent SVD are applied **only to the parameters targeted by LoRA**, not the entire model. To be specific, this is limited to the attention weights, which in the CLIP model amount to 768x768x4x12$\approx$28.31M parameters in the vision encoder and 512x512x4x12$\approx$12.58M in the text encoder. Combined, these account for only 27% of the model's 149.62M total parameters, significantly reducing the computational load. In terms of wall-clock time, the one-time construction of the initial principal subspace $W_p$ is negligible, taking only 0.02% of the total training time. For per-task overhead, the gradient-informed LoRA initialization (Algorithm 1, Line 4) accounts for 16.9% of the training time, while the subspace update via dominant feature extraction (Algorithm 1, Line 8) requires an additional 5.6% of the training time. Overall, this modest and controlled overhead is a justified trade-off for effectively balancing plasticity, backward stability, and forward stability.

---

> > ### Author Response · Authors · 2025-11-18
> > **Part II**
> >
> > > **W3**: Relies on crucial hyperparameters (e.g.,$\epsilon_w$ ,$\epsilon_f$ ) whose robustness and sensitivity are not deeply analyzed.
> >
> > Thank you for this valuable feedback. We had already included an empirical analysis for $\epsilon_{w(vision)}$ and $\epsilon_{w(text)}$ in Appendix C.2, which demonstrates its robustness within a reasonable range. We supplement the analysis for $\epsilon_f$, which controls the amount of information preserved for each past task.
> >
> > | $\epsilon_f$ | 0.9  | 0.95 | 0.99 | 0.995 |
> > | ------------ | ---- | ---- | ---- | ----- |
> > | Last         | 81.3 | 83.0 | 86.1 | 86.2  |
> >
> > The result shows that the performance improves as $\epsilon_f$ increases, stabilizing at high values $\geq$ 0.99, which indicates that preserving a high fraction of the feature energy is critical for maintaining the stability of previously learned tasks. Based on this, we set $\epsilon_f$=0.99 in our experiments as a robust choice.
> >
> > We further provide an analysis of how $\epsilon_f$ mitigates forgetting in the followings:
> >
> > Let $X \in \mathbb{R}^{d \times n}$ denote the layer feature matrix of a task. We define $M \in \mathbb{R}^{d \times p}$ as the top-$p$ principal components of $X$, which capture at least an $\epsilon_f$ fraction of the total variance, satisfying$\|\|M M^{\top} X \|\|_{F}^{2} \ge  \epsilon_f \|\| X \|\|_F^{2}$.
> >
> > The feature matrix $X$ can then be decomposed as: $ X = M M^{\top} X + (I-MM^{\top})X$,  where $\|\|(I-MM^{\top})X\|\|_{F} ^{2} = \|\| X\|\|_F^{2} -   \|\|M M^{\top} X \|\|_F^{2} \le (1-  \epsilon_f) \|\|X\|\|_F^{2}$ .
> >
> > Consider an incremental parameter update $\Delta W \in \mathbb{R}^{d \times d}$ when learning a new task. If we enforce that $\Delta W$ is orthogonal to the principal subspace, $\Delta W M = 0$, then: $ \Delta WX =   \Delta W \left(M M^{\top} X + (I-MM^{\top})X\right) = \Delta W (I-MM^{\top})X$.
> >
> > Therefore, the interference induced on the features satisfies: $\|\|\Delta W X\|\|_F^{2} = \|\|\Delta W (I-MM^{\top})X\|\|_F^{2} \le (1 - \epsilon_f)\|\|\Delta W \|\|_F^{2} \|\| X\|\|_F^{2}$.
> >
> > Since the residual $(I - MM^{\top})X$ contains at most $(1 - \epsilon_f)$ fraction of the total variance of $X$, the update $\Delta W$ only acts on low-variance directions. Consequently, the induced change $\|\|\Delta W X\|\|_F^2$ remains small, ensuring that the learned task representations are minimally perturbed, thereby mitigating catastrophic forgetting.
> >
> > Empirically, we obtain $M$ by performing SVD on the feature matrix $X = U\Sigma V^{\top}$. The principal directions of $X$ correspond to the left singular vectors associated with the largest singular values, whose squared magnitudes represent the variance captured by each component. We select the smallest $p$ such that the top-$p$ singular values account for at least an $\epsilon_f$ fraction of the total variance, i.e., $\sum_{i=1}^{p}\sigma_i^2 \geq \epsilon_f \sum_{i}\sigma_i^2$, and form $M$ by taking the first $p$ columns of $U$. This ensures that $MM^{\top}X$ retains at least an $\epsilon_f$ proportion of the energy in $X$.
> >
> >
> >
> > In addition to the points above, we would like to highlight that we have scaled KeepLoRA to the **LLaVA-1.5-7B** model for vision question answering tasks. **Please refer to our main Official Comment for these results, with further details available in the revised paper (Tables 3, 4 and Appendix B.1 and B.4).**

---

### Author Response · Authors · 2025-11-17
**Scale KeepLoRA to LLaVA-1.5-7b for vision question answering tasks**

To address the concern regarding the method's effectiveness on more complex VLM tasks, we have conducted extensive new experiments using **LLaVA-1.5-7b** on two recently proposed continual learning benchmarks **MLLM-DCL** and **UCIT**. These benchmarks specifically feature various instruction formats, including image captioning, visual question answering, and multiple-choice questions. Our results demonstrate that KeepLoRA achieves state-of-the-art performance across all evaluation metrics Transfer, Average, and Last scores. The training was performed on 4 × NVIDIA H100 GPUs for no more than 3 epochs. Further experimental details are provided in the revised paper. The code for these new experiments is available at: https://anonymous.4open.science/r/KeepLoRA_VQA-2BDA/

**MLLM-DCL**

| Method       | Sensing   | Medical   | Driving   | Science   | Finance   | Avg.      |
| ------------ | --------- | --------- | --------- | --------- | --------- | --------- |
| Zero-shot    | 32.29     | 28.28     | 15.59     | 35.55     | 62.56     |           |
| **Transfer** |           |           |           |           |           |           |
| LoRA-FT      | --        | 28.10     | 17.44     | 34.03     | 50.19     | 32.44     |
| O-LoRA       | --        | 28.37     | 18.37     | 33.72     | 52.53     | 33.25     |
| CL-MoE       | --        | 28.25     | 19.38     | 34.08     | 48.56     | 32.57     |
| SEFE         | --        | 28.10     | **19.63** | 33.85     | 52.36     | 33.49     |
| KeepLoRA     | --        | **28.49** | 16.63     | **34.13** | **55.61** | **33.71** |
| **Average**  |           |           |           |           |           |           |
| LoRA-FT      | 73.34     | 44.94     | 31.38     | 38.79     | 57.84     | 49.26     |
| O-LoRA       | 75.04     | 45.71     | 32.62     | 38.54     | 59.64     | 50.31     |
| CL-MoE       | 74.19     | 45.60     | 32.08     | 38.88     | 56.68     | 49.49     |
| SEFE         | 77.71     | 47.69     | 35.35     | 38.99     | 59.57     | 51.86     |
| KeepLoRA     | **79.55** | **50.80** | **37.53** | **40.70** | **62.35** | **54.19** |
| **Last**     |           |           |           |           |           |           |
| LoRA-FT      | 69.34     | 44.30     | 29.10     | 41.44     | 88.43     | 54.52     |
| O-LoRA       | 72.30     | 46.89     | 31.59     | 41.50     | 88.06     | 56.07     |
| CL-MoE       | 71.83     | 47.36     | 29.49     | 41.48     | 89.16     | 55.86     |
| SEFE         | 77.05     | 50.86     | 40.27     | 42.98     | 88.40     | 59.91     |
| KeepLoRA     | **78.76** | **54.34** | **50.19** | **49.48** | **89.30** | **64.41** |

**UCIT**

| Method       | ImgNet-R  | ArxivQA   | VizWiz    | IconQA    | CLEVR     | Flickr30k | Avg.      |
| ------------ | --------- | --------- | --------- | --------- | --------- | --------- | --------- |
| Zero-shot    | 16.27     | 53.73     | 38.39     | 19.20     | 20.63     | 41.88     |           |
| **Transfer** |           |           |           |           |           |           |           |
| LoRA-FT      | --        | 52.63     | 18.30     | 6.02      | 16.97     | 40.29     | 26.84     |
| O-LoRA       | --        | 52.87     | 19.57     | 4.42      | 16.85     | 41.04     | 26.95     |
| CL-MoE       | --        | 52.00     | 19.32     | 7.37      | 17.81     | 41.28     | 27.56     |
| SEFE         | --        | **53.33** | 18.68     | 7.48      | 17.03     | 40.90     | 27.48     |
| KeepLoRA     | --        | 52.83     | **20.39** | **9.18**  | **18.12** | **41.50** | **28.40** |
| **Average**  |           |           |           |           |           |           |           |
| LoRA-FT      | 75.98     | 77.78     | 41.56     | 38.83     | 34.56     | 43.25     | 51.99     |
| O-LoRA       | 82.43     | 80.06     | 41.73     | 35.87     | 33.94     | 43.74     | 52.96     |
| CL-MoE       | 80.16     | 77.10     | 40.43     | 30.33     | 33.10     | 43.95     | 50.85     |
| SEFE         | 85.49     | 78.55     | **42.92** | **40.33** | 34.80     | 43.64     | 54.29     |
| KeepLoRA     | **86.50** | **83.63** | 42.66     | 40.08     | **35.24** | **44.11** | **55.37** |
| **Last**     |           |           |           |           |           |           |           |
| LoRA-FT      | 58.60     | 76.73     | 45.72     | 67.43     | 61.57     | **58.03** | 61.35     |
| O-LoRA       | 74.17     | 80.93     | 45.30     | 62.87     | 63.83     | 57.24     | 64.06     |
| CL-MoE       | 67.17     | 75.77     | 44.38     | 52.63     | 54.40     | 57.28     | 58.61     |
| SEFE         | 80.23     | 79.13     | **47.11** | **69.40** | 65.70     | 57.33     | 66.48     |
| KeepLoRA     | **82.43** | **86.70** | 46.54     | 67.80     | **66.40** | 57.18     | **67.84** |

---

### Comment · Area_Chair_FVfM · 2025-11-25
**Discussion Period**

Dear Reviewers and Authors,

Thank you to the authors for submitting your rebuttal. We kindly encourage reviewers to take a moment to read the response and share any follow-up thoughts. Your timely engagement at this stage is highly valuable and helps ensure a fair, well-informed final decision.

We appreciate everyone’s efforts and contributions to the process.

Warm regards, Your AC

---

### Author Response · Authors · 2025-12-01
**Summary of Rebuttal**

Dear AC,

We thank all reviewers for their valuable and constructive feedback and briefly summarize the author-reviewer discussion progress as follows.

Reviewers explicitly acknowledged the value of our work in the following aspects:

   - **Good Motivation:** (zp4o S1, FhjQ S1, gbxX S1).

   - **Extensive Analysis & SOTA Performance:** (zp4o S3, FhjQ S3, 1c8k S1, gbxX S4).

   - **Solid Theory:** (zp4o S2, FhjQ S3, 1c8k S3, gbxX S2).

In our responses, we have carefully addressed all concerns raised by reviewers, with a focus on scalability and robustness:

   - **Scalability (1c8k, gbxX):** In addition to image classification tasks, **we perform evaluations of our approach on two VQA bechmark datasets based on LLaVA-1.5-7b model**, which demonstrates the effectiveness of the method on more complex, language-heavy VLM tasks.

   - **Sensitivity & Theory (zp4o, FhjQ):** We conducted comprehensive sensitivity experiments complemented by deeper theoretical explanations to verify the method's robustness.

   - **Clarifications:** We also provided detailed explanations for other questions (e.g., orthogonality constraints), which were well-received.

During rebuttal, **our ratings have improved from 6/6/4/6 to 6/6/6/6**. Specifically, Reviewer 1c8k raised the initial score from 4 to 6, and Reviewer gbxX decided to keep the positive rating. Reviewers zp4o and FhjQ gave positive initial ratings but have not provided further replies.

We sincerely appreciate your time and effort in handling our paper, and we hope our brief summarization of the rebuttal process can assist your assessment.

Best regards,

The Authors

---

### Meta-Review · Area_Chair_5E2j · 2025-12-29

**Summary:**

The paper introduces KeepLoRA, a method for continual learning in pre-trained vision-language models. It utilizes a subspace analysis to separate general knowledge and task-specific knowledge. By restricting LoRA updates to the residual subspace, KeepLoRA effectively maintains plasticity, backward stability, and forward stability. The authors provide both theoretical justifications and empirical validation, demonstrating the method’s state-of-the-art performance on the MTIL benchmark. Several concerns were raised during the review process, particularly about the scalability of the approach, the potential interference between similar tasks in the residual subspace, and the sensitivity of hyperparameters.

**Reviewer Concerns:**

The authors addressed the reviewers' primary concerns effectively. In particular, they clarified the scalability of the method by providing additional experiments using the LLaVA-1.5-7B model on complex vision-language tasks like vision question answering. Sensitivity to hyperparameters was thoroughly analyzed, and the authors also conducted sensitivity experiments for key parameters such as epsilon_w and r. However, some concerns remain regarding the potential for interference between highly similar tasks, especially with fine-grained classification tasks like StanfordCars and Aircraft. The paper could have further explored the computational and memory overhead of the method, particularly in larger models. Despite these issues, the responses and the new results presented in the rebuttal were well-received by the reviewers.

**Reviewer Scores:**

Reviewers initially gave scores ranging from 4 to 6, but after reading the rebuttal and the new experimental results, all reviewers maintained or slightly adjusted their scores. Reviewer 1c8k raised their score from 4 to 6, citing the additional experiments as crucial for confirming the method’s effectiveness on more complex tasks. Reviewer gbxX kept their positive rating, indicating that the authors had addressed their concerns sufficiently. Reviewer zp4o and FhjQ did not adjust their initial positive ratings, reflecting general satisfaction with the rebuttal.

---

### Decision · Program_Chairs · 2026-01-26

Accept (Poster)